# Data Augmentation is a Hyperparameter:

## Cherry-picked Self-Supervision for Unsupervised Anomaly Detection is Creating the Illusion of Success

**Jaemin Yoo**                                                  *jaeminyoo@cmu.edu*
*Heinz College of Information Systems and Public Policy*
*Carnegie Mellon University*

**Tiancheng Zhao**                                              *tianchen@andrew.cmu.edu*
*School of Architecture*
*Carnegie Mellon University*

**Leman Akoglu**                                                *lakoglu@andrew.cmu.edu*
*Heinz College of Information Systems and Public Policy*
*Carnegie Mellon University*

**Reviewed on OpenReview:** *https://openreview.net/forum?id=HyzCuCV1jH*

## Abstract

Self-supervised learning (SSL) has emerged as a promising alternative to create supervisory signals to real-world problems, avoiding the extensive cost of manual labeling. SSL is particularly attractive for unsupervised tasks such as anomaly detection (AD), where labeled anomalies are rare or often nonexistent. A large catalog of augmentation functions has been used for SSL-based AD (SSAD) on image data, and recent works have reported that the type of augmentation has a significant impact on accuracy. Motivated by those, this work sets out to put image-based SSAD under a larger lens and investigate the role of data augmentation in SSAD. Through extensive experiments on 3 different detector models and across 420 AD tasks, we provide comprehensive numerical and visual evidences that the alignment between data augmentation and anomaly-generating mechanism is the key to the success of SSAD, and in the lack thereof, SSL may even impair accuracy. To the best of our knowledge, this is the first meta-analysis on the role of data augmentation in SSAD.

## 1 Introduction

Machine learning has made tremendous progress in creating models that can learn from labeled data. However, the cost of high-quality labeled data is a major bottleneck for the future of supervised learning. Most recently, self-supervised learning (SSL) has emerged as a promising alternative; in essence, SSL transforms an unsupervised task into a supervised one by self-generating labeled examples. This new paradigm has had great success in advancing NLP (Devlin et al., 2019; Conneau et al., 2020; Brown et al., 2020) and has also helped excel at various computer vision tasks (Goyal et al., 2021; Ramesh et al., 2021; He et al., 2022). Today, SSL is arguably the key toward "unlocking the dark matter of intelligence" (LeCun & Misra, 2021).

**SSL for unsupervised anomaly detection.** SSL is particularly attractive for *unsupervised* tasks such as anomaly detection (AD), where labeled data is either rare or nonexistent, costly to obtain, or nontrivial to simulate in the face of unknown anomalies. Thus, the literature has seen a recent surge of SSL-based AD (SSAD) techniques (Golan & El-Yaniv, 2018; Bergman & Hoshen, 2020; Li et al., 2021; Sehwag et al., 2021; Cheng et al., 2021; Qiu et al., 2021). The common approach they take is incorporating self-generated *pseudo* anomalies into training, which aims to separate those from the inliers. The pseudo-anomalies are often created

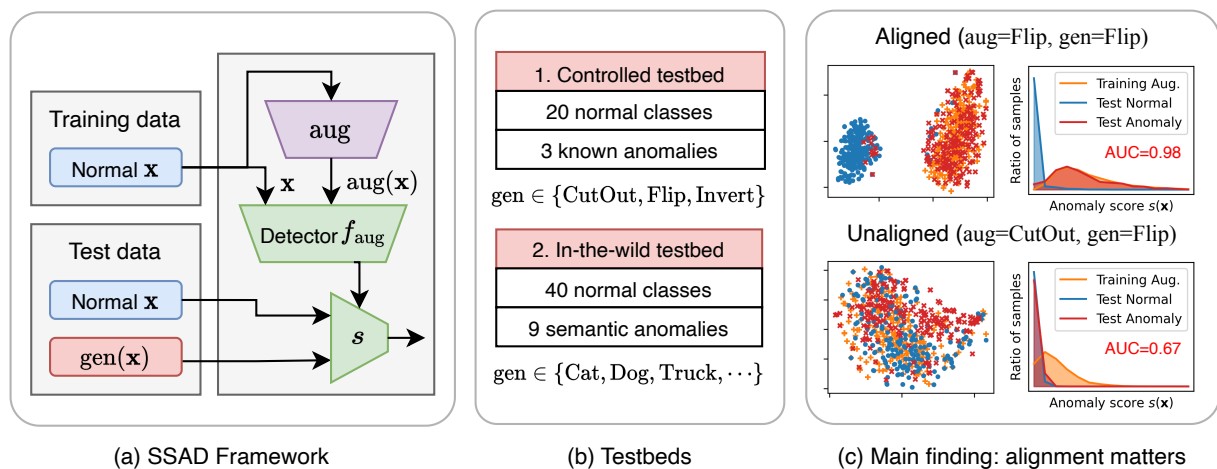

Figure 1: (best in color) A visual summary of our work. (a) In SSAD, we train a detector $f_{\mathsf{aug}}$ from normal data $\mathbf{x}$ and pseudo anomalies $\mathsf{aug}(\mathbf{x})$ generated from an augmentation function $\mathsf{aug}$, and feed the output of $f_{\mathsf{aug}}$ to a score function $s$. We denote the underlying anomaly-generating mechanism in test data by $\mathsf{gen}$. (b) We use two testbeds in experiments: 1) $20 \times 3$ tasks with known (or controlled) anomalies, and 2) $40 \times 9$ tasks for real-world semantic class anomalies. (c) We illustrate two cases with and without alignment between $\mathsf{aug}$ and $\mathsf{gen}$: (left) data embeddings and (right) anomaly score histograms. SSAD succeeds when $\mathsf{aug}$ mimics $\mathsf{gen}$, showing almost perfect AUC (top), while it fails when $\mathsf{aug}$ is misaligned with $\mathsf{gen}$ (bottom).

in one of two ways: (*i*) by transforming inliers through data augmentation[1] or (*ii*) by "outlier-exposing" the training to external data sources (Hendrycks et al., 2019; Ding et al., 2022a). The former synthesizes artificial data samples, while the latter uses existing real-world samples from external data repositories.

While perhaps re-branding under the name SSL, the idea of injecting artificial anomalies to inlier data to create a labeled training set for AD dates back to early 2000s (Abe et al., 2006; Steinwart et al., 2005; Theiler & Cai, 2003). Fundamentally, under the uninformative *uniform* prior for the (*unknown*) anomaly-generating distribution, these methods are asymptotically consistent density level set estimators for the support of the inlier data distribution (Steinwart et al., 2005). Unfortunately, they are ineffective and sample-inefficient in high dimensions as they require a massive number of sampled anomalies to properly fill the sample space.

**SSL-based AD incurs hyperparameters to choose.** With today's SSL methods for AD, we observe a shift toward various *non-uniform* priors on the distribution of anomalies. In fact, current literature on SSAD is laden with many different aforementioned[1] forms of generating pseudo anomalies, each introducing its own inductive bias. While this offers a means for incorporating domain expertise to detect known types of anomalies, in general, anomalies are hard to define apriori or one is interested to detect unknown anomalies. As a consequence, the success on any AD task depends on *which augmentation* function is used or *which external dataset* the learning is exposed to as pseudo anomalies. In other words, SSL calls forth *hyperparameter* to choose carefully. The supervised ML community has demonstrated that different downstream tasks benefit from different augmentation strategies and associated invariances (Ericsson et al., 2022), and thus integrates these "*data augmentation hyperparameters*" into model selection (MacKay et al., 2019; Zoph et al., 2020; Ottoni et al., 2023; Cubuk et al., 2019), whereas problematically, the AD community appears to have turned a blind eye to the issue. Tuning hyperparameters without any labeled data is admittedly challenging, however, *unsupervised AD does not legitimize cherry-picking critical SSL hyperparameters, which creates the illusion that SSL is the "magic stick" for AD and over-represents the level of true progress in the field.*

**Evidence from the literature and simulations.** Let us provide illustrative examples from the literature as well as our own simulations showing that SSL hyperparameters have significant impact on AD performance. Golan & El-Yaniv (2018) have found that geometric transformations (e.g. rotation) outperform pixel-level augmentation for detecting semantic anomalies (e.g. airplanes vs. birds). In contrast, Li et al. (2021) have

---

[1]E.g., rotation, blurring, masking, color jittering, CutPaste (Li et al., 2021), as well as "cocktail" augmentations like GEOM (Golan & El-Yaniv, 2018) and SimCLR (Chen et al., 2020).

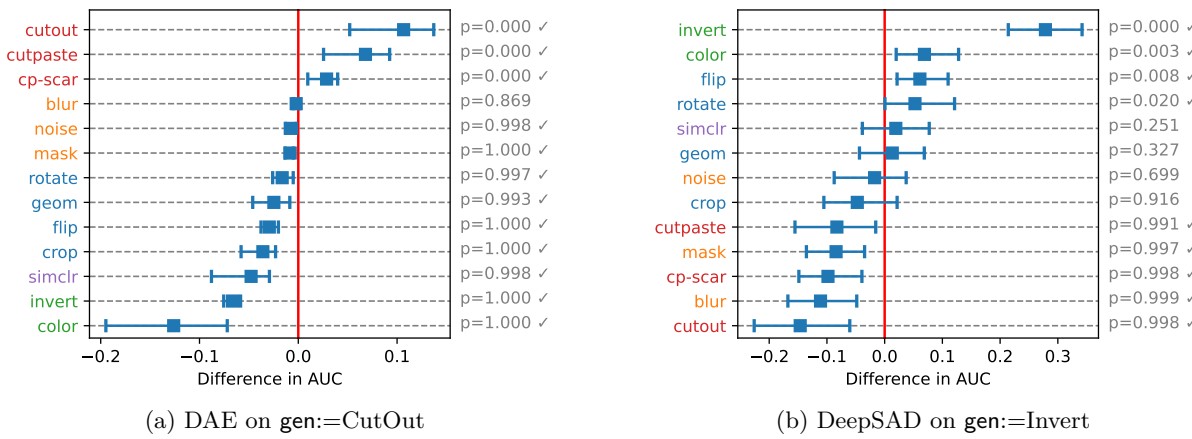

(a) DAE on `gen:=`CutOut

(b) DeepSAD on `gen:=`Invert

Figure 2: (best in color) Relative AUC (x-axis) of SSL-based DAE and DeepSAD in comparison to their un-supervised counterparts, respectively AE and DeepSVDD, on the controlled testbed with different anomaly-generating function `gen` in (a) vs. (b). Color in augmentation names (y-axis) depicts their categories. (a) Local augmentations (in red) perform well thanks to the high alignment with `gen:=`CutOut, while others (e.g. in green) can even hurt the accuracy significantly (*p*-values in gray). (b) Similarly, color-based augmentations (in green) improve the baseline AUC significantly since they agree with `gen:=`Invert, while several others even impair the accuracy. See our Finding 1 in Sec. 4 for further details.

shown that such global transformations perform significantly poorly at detecting small defects in industrial object images (e.g. scratched wood, cracked glass, etc.), and thus proceeding to propose local augmentations such as cut-and-paste, which performed significantly better than rotation (90.9 vs. 73.1 AUC on average; see their Table 1). We replicate these observations in our simulations; Fig. 2 shows that local augmentation functions (in red) improve performance over the unsupervised detector when test anomalies are simulated to be local (mimicking small industrial defects), while other choices may even worsen the performance (!).

On the other hand, Ding et al. (2022a) consider three different augmentation functions as well as two external data sources for outlier-exposure (OE), with significant differences in performance (see their Table 4). They have compared to baseline methods by picking CutMix augmentation, the best one in their experiment, on all but three medical datasets which instead use OE from another medical dataset. Similarly, in evaluating OE-based AD (Hendrycks et al., 2019; Liznerski et al., 2022), the authors have picked different external datasets to training depending on the target/test dataset; specifically, they use as OE data the 80 Million Tiny Images (superset of CIFAR-10) to evaluate on CIFAR-10, whereas they use ImageNet-22K (superset of ImageNet-1K) to evaluate on ImageNet. Both studies provide evidence that the choice of self-supervision matters, yet raise concerns regarding fair evaluation and comparison to baselines.

As we argue later in our study, the underlying driver of success for SSAD is that the more the *pseudo* anomalies mimic the type/nature of the *true* anomalies in the test data, the better the AD performance. This is perhaps what one would expect, i.e. is unsurprising, yet it is essential to emphasize that true anomalies are *unknown* at training time. Any particular choice of generating the pseudo anomalies inadvertently leads to an inductive bias, ultimately yielding biased outcomes. This phenomenon was recently showcased through an eye-opening study (Ye et al., 2021); a detector becomes biased when pseudo anomalies are sampled from a biased subset of true anomalies, where the test error is lower on the known/sampled type of anomalies, at the expense of larger errors on the unknown anomalies—even when they are easily detected by an *unsupervised* detector. We also replicate their findings in our simulations. Fig. 3a shows that augmenting the inlier Airplane images by rotation leads to improved detection by the self-supervised DeepSAD on average; but with different performance distribution across individual anomaly classes: compared to (unsupervised) DeepSVDD, it better detects Bird, Frog, and Deer images as anomalies, yet falters in detecting Automobile and Truck—despite that these latter are easier to detect by the unsupervised DeepSVDD (!). Results in Fig. 3b for DAE are similar, also showcasing the biased detection outcomes under this specific self-supervision.

**Our study and contributions.** While the aforementioned studies serve as partial evidence, the current literature lacks systematic scrutiny of the working assumptions for the success of SSL for AD. Importantly,

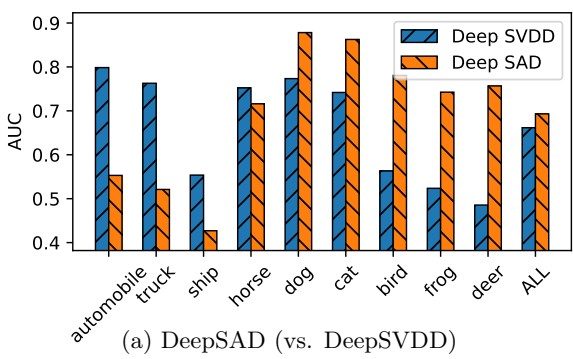 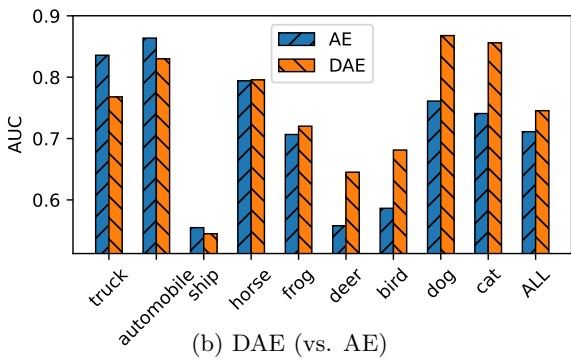

(a) DeepSAD (vs. DeepSVDD)          (b) DAE (vs. AE)

Figure 3: AUC performances on different classes of semantic anomalies, comparing SSL-based DeepSAD and DAE with their unsupervised counterparts, respectively DeepSVDD and AE, on CIFAR-10. We use Airplane as the inlier class and Rotate as the augmentation function. Self-supervision improves accuracy on average (last column) but impairs it on certain individual anomaly classes that are easily detected by unsupervised models. This shows that the negative effect of biased supervision as observed by Ye et al. (2021) also applies to SSAD, leading to biased detection outcomes across anomaly types. See Obs. 3 in Sec. 4 for details.

hyperparameter selection and other design choices for SSAD are left unjustified in recent works, and many of them even make different choices for different datasets in the same paper, violating the real-world setting of unsupervised AD. In this work, we set out to put SSAD under a larger investigative lens. As discussed, Fig.s 2 and 3 give clear motivation for our research; comparing SSL-based models DAE and DeepSAD with their *unsupervised* counterparts, respectively AE and DeepSVDD, showcases the importance of the choice of self-supervised augmentation. Toward a deeper understanding, we run extensive experiments on 3 detector models over 420 different AD tasks, studying why and how self-supervision succeeds or otherwise fails. We show the visual summary of our work in Fig. 1, including our main finding on the alignment.

Our goal is to uncover pitfalls and bring clarity to the growing field of SSL for AD, rather than proposing yet-another SSAD model. To the best of our knowledge, this is the first meta-analysis study on SSAD with carefully designed experiments and statistical tests. Our work is akin to similar investigative studies on other aspects of deep learning (Erhan et al., 2010; Ruff et al., 2020a; Mesquita et al., 2020) that aim to critically review and understand trending research areas. We expect that our work will provide a better understanding of the role of SSL for AD and help steer the future directions of research on the topic. We summarize our main contributions as follows:

- **Comprehensive study of SSL on image AD:** Our study sets off to answer the following questions: How do different choices for pseudo anomaly generation impact detection performance? Can data augmentation *degrade* performance? What is the key contributor to the success or the failure of SSL on image AD? To this end, we conduct extensive experiments on controlled as well as in-the-wild testbeds using 3 different types of SSAD models across 420 different AD tasks (see Fig. 1).

- **Alignment between pseudo and true anomalies:** Our study presents comprehensive evidence that augmentation remains to be a hyperparameter for SSAD—the choice of which heavily influences performance. The "X factor" is the alignment between the augmentation function aug and the true anomaly-generating function gen, where alarmingly, poor alignment even hurts performance (see Fig. 2) or leads to a biased error distribution (see Fig. 3). Bottom line is that effective hyperparameter selection is essential for unlocking the game-changing potential of SSL for AD.

For reproducibility, we open-source our implementations at https://github.com/jaeminyoo/SSL-AD.

## 2 Related Work and Preliminaries

### 2.1 Related Work on Self-supervised Anomaly Detection (SSAD)

**Generative Models.** Generative models learn the distribution of normal (i.e. inlier) samples and measure the anomaly score of an example based on its distance from the learned distribution. Generative models for

AD include autoencoder-based models (Zhou & Paffenroth, 2017; Zong et al., 2018), generative adversarial networks (Akcay et al., 2018; Zenati et al., 2018), and flow-based models (Rudolph et al., 2022). Recent works (Cheng et al., 2021; Ye et al., 2022) proposed denoising autoencoder (DAE)-based models for SSAD by adopting domain-specific augmentation functions instead of traditional Gaussian noise or Bernoulli masking (Vincent et al., 2008; 2010). SSAD addresses the limitation of generative models, where out-of-distribution data exhibit higher likelihood than the training data itself (Theis et al., 2015; Nalisnick et al., 2018).

**Classifier-based Models.** Augmentation prediction is to learn a classifier by creating pseudo labels of samples from multiple augmentation functions. The classifier trained to differentiate augmentation functions is used to generate better representations of data. Many augmentation functions were proposed for AD in this sense, mostly for image data, including geometric transformation (Golan & El-Yaniv, 2018), random affine transformation (Bergman & Hoshen, 2020), local image transformation (Li et al., 2021), and learnable neural network-based transformation (Qiu et al., 2021). Outlier exposure (OE) (Ding et al., 2022a; Hendrycks et al., 2019) uses an auxiliary dataset as pseudo anomalies for training a classifier that separates it from normal data. Since the choice of an auxiliary dataset has a large impact on the performance, previous works have chosen a suitable dataset for each target task considering the nature of true anomalies.

**Semi-supervised Models.** Semi-supervised models assume that a few samples of true anomalies are given at training (Ruff et al., 2020b; Ye et al., 2021). A recent work (Ye et al., 2021) has shown that observing a biased subset of anomalies induces a bias also in the model's predictions, impairing its accuracy on unseen types of anomalies even when they are easy to detect by unsupervised models. Motivated by this, we pose semi-supervised AD as a form of self-supervised learning in that limited observations of (pseudo) anomalies and its proper alignment with the distribution of true anomalies play an essential role in the performance.

## 2.2 Related Work on Self-supervised Learning and Automating Augmentation

While our work focuses on SSL for anomaly detection problems in the absence of labeled anomalies to learn from, we also outline related work on SSL in other areas more broadly.

**Self-supervised Learning.** Self-supervised learning (SSL) has seen a surge of attention for pre-training foundation models (Bommasani et al., 2021) like large language models that can generate remarkable human-like text (Zhou et al., 2023). Self-supervised representation learning has also offered an astonishing boost to a variety of tasks in natural language processing, vision, and recommender systems (Liu et al., 2021). SSL has been argued as the key toward "unlocking the dark matter of intelligence" (LeCun & Misra, 2021).

**Automating Augmentation.** Recent work in computer vision (CV) has shown that the success of SSL relies heavily on well-designed data augmentation strategies (Tian et al., 2020; Steiner et al., 2022; Touvron et al., 2022; Ericsson et al., 2022) as in a recent study on the semantic alignment of datasets in semi-supervised learning (Calderon-Ramirez et al., 2022). Although the CV community proposed approaches for automating augmentation (Cubuk et al., 2019; 2020), such approaches are not applicable to SSAD, where labeled data are not available at the training. While augmentation in CV plays a key role in improving generalization by accounting for invariances (e.g. mirror reflection of a dog is still a dog), augmentation in SSAD presents the classifier with specific kinds of pseudo anomalies, solely determining the decision boundary. Our work is the first attempt to rigorously study the role and effect of augmentation in SSAD.

## 2.3 Related Work on Unsupervised Outlier Model Selection

Through this work, we show that data augmentation for SSAD is yet-another hyperparameter and thus more broadly relates to the problem of unsupervised (outlier) model selection. Unsupervised hyperparameter tuning (i.e. model selection) is nontrivial in the absence of labeled data (Ma et al., 2023), where the literature is recently growing (Zhao et al., 2021; 2022; Zhao & Akoglu, 2022; Ding et al., 2022b; Zhao et al., 2023). Existing works can be categorized into two groups depending on how they estimate the performance of an outlier detection model without using any labeled data. The first group uses internal performance measures that are based solely on the model's output and/or input data (Ma et al., 2023). The second group consists of meta-learning-based approaches (Zhao et al., 2021; Zhao & Akoglu, 2022), which facilitate model selection for a new unsupervised task by leveraging knowledge from similar historical tasks.

### 2.4 Problem Definition and Notation

We define the anomaly detection problem (Li et al., 2021; Qiu et al., 2021) as follows:

- **Given:** Set $\mathcal{D} = \{\mathbf{x}_i\}_{i=1}^N$ of normal data, where $N$ is the number of training samples, and $\mathbf{x}_i \in \mathbb{R}^d$;
- **Find:** Score function $s(\cdot) \in \mathbb{R}^d \to \mathbb{R}$ such that $s(\mathbf{x}) < s(\mathbf{x}')$ if $\mathbf{x}$ is normal and $\mathbf{x}'$ is abnormal.

The definition of normality (or abnormality) is different across datasets. For the generality of notations, we introduce an anomaly-generating function $\mathsf{gen}(\cdot) : \mathbb{R}^d \to \mathbb{R}^d$ that creates anomalies from normal data. We denote a test set that contains both normal data and anomalies generated by $\mathsf{gen}(\cdot)$, as $\mathcal{D}_{\mathsf{gen}}$.

The main challenge of anomaly detection is the lack of labeled anomalies in the training set. Self-supervised anomaly detection (SSAD) addresses the problem by generating pseudo-anomalies through a data augmentation function $\mathsf{aug}(\cdot) \in \mathbb{R}^d \to \mathbb{R}^d$. SSAD trains a neural network $f(\cdot; \theta) \in \mathbb{R}^d \to \mathbb{R}^h$ using $\mathsf{aug}(\cdot)$, where $\theta$ is the set of parameters, and defines a score function $s(\cdot)$ on top of the data representations learned by $f$. We denote a network $f$ trained with $\mathsf{aug}$ by $f_{\mathsf{aug}}$ when there is no ambiguity.

### 2.5 Representative Models for SSAD

The main components of an SSAD model are an objective function $l$ and an anomaly score function $s$. These determine how to utilize $\mathsf{aug}$ in the training of $f_{\mathsf{aug}}$ and how to quantify anomalies. We introduce three models from the three categories in Sec. 2, respectively, focusing on their definitions of $l$ and $s$.

**Denoising Autoencoder (DAE).** The objective function for DAE (Vincent et al., 2008) is given as $l(\theta) = \sum_{\mathbf{x} \in \mathcal{D}} \|f(\mathsf{aug}(\mathbf{x}); \theta) - \mathbf{x}\|_2^2$. That is, $f_{\mathsf{aug}}$ aims to reconstruct the original $\mathbf{x}$ from $\mathsf{aug}(\mathbf{x})$. We use the mean squared error between $\mathbf{x}$ and the reconstructed version $f(\mathbf{x}; \theta)$ as an anomaly score. The training of a *vanilla* (no-SSL) autoencoder (AE) is done by employing the identity function $\mathsf{aug}(\mathbf{x}) = \mathbf{x}$.

**Augmentation Predictor (AP).** The objective function for AP (Golan & El-Yaniv, 2018) is given as $l(\theta) = \sum_{\mathbf{x} \in \mathcal{D}} \sum_{k=1}^K \mathrm{NLL}(f(\mathsf{aug}_k(\mathbf{x}); \theta), k)$, where $K$ is the number of separable *classes* of $\mathsf{aug}$, $\mathsf{aug}_k$ is the $k$-th class of $\mathsf{aug}$ which is an augmentation function itself, and $\mathrm{NLL}(\hat{\mathbf{y}}, y) = -\log \hat{\mathbf{y}}_y$ is the negative loglikelihood. The idea is to train a $K$-class classifier that can predict the class of augmentation used to generate $\mathsf{aug}_k(\mathbf{x})$. The separable classes are defined differently for each $\mathsf{aug}$. For example, Golan & El-Yaniv (2018) sets $K = 4$ when $f_{\mathsf{aug}}$ is an image rotation function and sets each $\mathsf{aug}_k$ to the $e$-degree rotation with $e \in \{0, 90, 180, 270\}$. Unlike DAE and DeepSAD, there is no vanilla model of AP that works without $\mathsf{aug}$. The anomaly score $s(\mathbf{x})$ is computed as $s(\mathbf{x}) = -\sum_{k=1}^K [f(\mathsf{aug}_k(\mathbf{x}))]_k$, such that $\mathbf{x}$ receives a high score if its classification is failed.

**DeepSAD.** The objective function for DeepSAD (Ruff et al., 2020b) is given as $l(\theta) = \sum_{\mathbf{x} \in \mathcal{D}} \|f(\mathbf{x}; \theta) - \mathbf{c}\| + (\|f(\mathsf{aug}(\mathbf{x}); \theta) - \mathbf{c}\|)^{-1}$, where the hypersphere center $\mathbf{c}$ is set as the mean of the outputs obtained from an initial forward pass of the training data $\mathcal{D}$. We then re-compute $\mathbf{c}$ every time the model is updated during the training. The anomaly score $s(\mathbf{x})$ is defined as $\|\mathbf{x} - \mathbf{c}\|_2^2$, which is the squared distance between $\mathbf{x}$ and $\mathbf{c}$. We adopt DeepSVDD (Ruff et al., 2018) as the no-SSL vanilla version of DeepSAD by using only the first term of the objective function; it only makes the representations of data close to the hypersphere center.

## 3 Experimental Setup

**Models.** We run experiments on the 3 SSL-based detector models introduced in Sec. 2.5: DAE, DeepSAD, and AP. We also include the no-SSL baselines AE and DeepSVDD for DAE and DeepSAD, respectively, with the same model architectures. Details on the models are given in Appendix A.

**Augmentation Functions.** We study various types of augmentation functions, which are categorized into five groups. Bullet colors are the same as in Fig.s 2, 4, and 5.

- Geometric: Crop (Chen et al., 2020), Rotate, Flip, and GEOM (Golan & El-Yaniv, 2018).
- Local: CutOut (Devries & Taylor, 2017), CutPaste and CutPaste-scar (Li et al., 2021).
- Elementwise: Blur (Chen et al., 2020), Noise, and Mask (Vincent et al., 2010).
- Color-based: Invert and Color (jittering) (Chen et al., 2020).
- Mixed ("cocktail"): SimCLR (Chen et al., 2020).

Table 1: AUC of three detector models on CIFAR-10C and CIFAR-10; the first three `gen` functions are from CIFAR-10C, where Cut. means CutOut, while Sem. means the anomalies of different semantic classes in CIFAR-10. Every model performs best when `aug` and `gen` are matched, supporting Finding 1. Different `aug` functions take the first place in Semantic, where the alignment with `gen` is unknown.

| Augment | Anomaly-generating function (= `gen`) | | | | | | | | | | | |
| (= `aug`) | DAE | | | | DeepSAD | | | | AP | | | |
| | Cut. | Flip | Invert | Sem. | Cut. | Flip | Invert | Sem. | Cut. | Flip | Invert | Sem. |
| CutOut | **0.974** | 0.593 | 0.654 | 0.604 | **0.999** | 0.580 | 0.556 | 0.592 | **0.797** | 0.503 | 0.508 | 0.505 |
| Flip | 0.771 | **0.865** | 0.772 | **0.691** | 0.727 | **0.876** | 0.731 | 0.691 | 0.639 | **0.959** | 0.836 | 0.780 |
| Invert | 0.746 | 0.663 | **0.970** | 0.690 | 0.674 | 0.659 | **0.973** | **0.695** | 0.645 | 0.717 | **0.994** | 0.753 |
| GEOM | 0.813 | 0.726 | 0.724 | 0.621 | 0.938 | 0.758 | 0.699 | 0.682 | 0.760 | 0.944 | 0.881 | **0.863** |

Geometric functions make global geometric changes to the input images. Local augmentations, in contrast, modify only a part of an image such as by erasing a small patch. Elementwise augmentations modify every pixel individually. Color-based augmentations change the color of pixels, while mixed augmentations combine multiple categories of augmentation functions. Detailed information is given in Appendix B.

**Datasets.** Our experiments are conducted on two kinds of testbeds, containing 420 different tasks overall. The first is *in-the-wild* testbed, where one semantic class is selected as normal and another class is selected as anomalous. We include four datasets in the testbed: MNIST (Garris, 1994; LeCun et al., 1998), FashionMNIST (Xiao et al., 2017), SVHN (Netzer et al., 2011), and CIFAR-10 (Krizhevsky et al., 2009). Since each dataset has 10 different classes, we have 90 tasks for all possible pairs of classes for each dataset. The second is *controlled* testbed, where we adopt a known function as the anomaly-generating function `gen` to have full control of the anomalies. Given two datasets SVHN and CIFAR-10, we use three `aug` functions as `gen`: CutOut, Flip, and Invert, making 30 different tasks (10 classes × 3 anomalies) for each dataset. We denote these datasets by SVHN-C and CIFAR-10C, respectively.

**Evaluation.** Given a detector model $f$ and a test set $\mathcal{D}_{\text{gen}}$ containing both normal data and anomalies for each task, we compute the anomaly score $s(\mathbf{x})$ for each $\mathbf{x} \in \mathcal{D}_{\text{gen}}$. Then, we measure the ranking performance by the area under the ROC curve (AUC), which has been widely used for anomaly detection.

The extent of alignment is determined by the functional similarity between `aug` and `gen`. They are perfectly aligned if they are exactly the same function, and still highly aligned if they are in the same family, such as `aug`:=Rotate and `gen`:=Flip – both of which are geometric augmentations.

# 4 Success and Failure of Augmentation

Our experiments are geared toward studying two aspects of augmentation in SSAD: *when* augmentation succeeds or otherwise fails (in Sec. 4) and *how* augmentation works when it succeeds or fails (in Sec. 5). We first investigate when augmentation succeeds by comparing a variety of augmentation functions.

## 4.1 Main Finding: Augmentation is a Hyperparameter

We introduce the main finding on the relationship between `aug` and `gen` for the performance of SSAD, which we demonstrate through extensive experiments presented in Fig. 2 and Table 1.

**Finding 1.** *Let $\mathcal{D}_{\text{gen}}$ be a test set with the anomaly-generating function `gen`, and $f_{\text{aug}}$, $f_{\text{aug}'}$, and $f$ be detector models with `aug`, `aug`′, and without augmentation, respectively. Then, (i) $f_{\text{aug}}$ surpasses $f_{\text{aug}'}$ if `aug` is better aligned with `gen` than `aug`′ is, and (ii) $f$ surpasses $f_{\text{aug}}$ if the alignment between `aug` and `gen` is poor.*

In Fig. 2, we compare the performances of DAE and DeepSAD with their no-SSL baselines, AE and DeepSVDD, respectively. Recall that AP has no such vanilla baseline. We run the paired Wilcoxon signed-rank test (Groggel, 2000) between $f_{\text{aug}}$ and $f$. Each row summarizes the AUCs from 20 different tasks across two datasets (CIFAR-10C and SVHN-C), ten classes each. We report the (pseudo) medians, 95% confidence intervals, and $p$-values for each experiment. The $x$-axis depicts the relative AUC compared with that of $f$. We consider `aug` to be helpful (or harmful) if the $p$-value is smaller than 0.05 (or larger than 0.95).

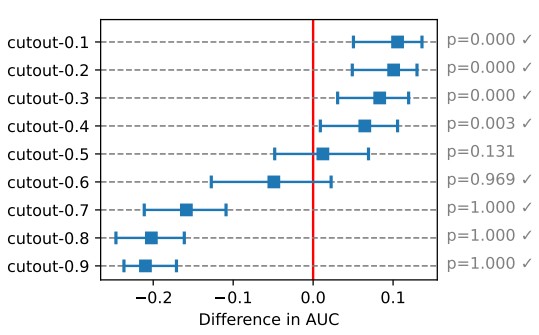

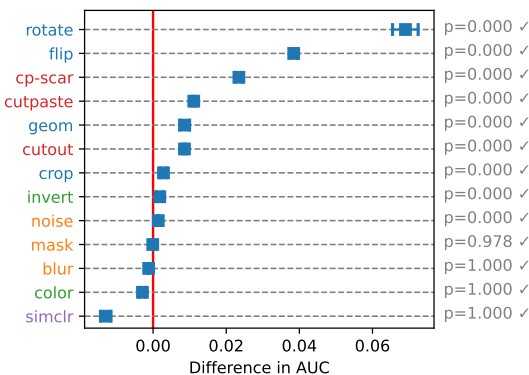

(a) Different `aug` hyperparameters

(b) Different model hyperparameters

Figure 4: (best in color) Relative AUC of DAE with different hyperparameters. (a) Controlled testbed with `aug`:=CutOut-$c$ where $c \in [0.1, 0.9]$ depicts the size of removed patches. Anomaly function `gen`:=CutOut sets the average patch size to 0.19. (b) In-the-wild testbed summarizing 16 different hyperparameter settings of DAE (and AE); each row is generated from 5760 values (360 tasks × 16 models). See Obs. 1 and 2.

In Fig.s 2a and 2b, `aug` functions with the best alignment with `gen` significantly improve the accuracy of $f$, supporting Finding 1: the local augmentations in Fig. 2a, and the color-based ones in Fig. 2b. In contrast, the remaining `aug` functions make negligible changes or even cause a significant decrease in accuracy. That is, the alignment between `aug` and `gen` determines the success or the failure of $f_{\text{aug}}$ on $\mathcal{D}_{\text{gen}}$. Our observations are consistent with other choices of `gen` functions as shown in Appendix D.

In Table 1, we measure the accuracy of all three models on CIFAR-10C and CIFAR-10 for multiple combinations of `aug` and `gen` functions. In the controlled tasks, `aug` = `gen` performs best in all three models, even though their absolute performances are different. On the other hand, different `aug` functions work best for the semantic anomalies: Flip for DAE, GEOM for AP, and Invert for DeepSAD. This is because different semantic classes are hard to be represented by a single `aug` function, making no `aug` achieve the perfect alignment with `gen`. In this case, different patterns are observed based on models: DAE shows similar accuracy with all `aug`, while AP shows clear strength with GEOM, as shown also in (Golan & El-Yaniv, 2018).

Our Finding 1 may read obvious,[2] but has strong implications for selecting testbeds for fair and accurate evaluation of existing work. The literature does not concretely state, recognize, or acknowledge the importance of alignment between `aug` and `gen`, even though it determines the success of a given framework. Our study is the first to make the connection explicit and provide quantitative results through extensive experiments. Moreover, we conduct diverse types of qualitative and visual inspections on the effect of data augmentation, further enhancing our understanding in various aspects (discussed later in Sec. 4.3 and 5).

## 4.2 Continuous Augmentation Hyperparameters

Next, we show through experiments that Finding 1 is consistent with augmentation functions with different continuous hyperparameters, and robust to different choices of model hyperparameters.

**Observation 1.** *The alignment between* `aug` *and* `gen` *is determined not only by the functional type of* `aug`, *but also by the amount of modification made by* `aug`, *which is determined by its continuous hyperparameter(s).*

Fig. 4a compares the effect of `aug`:=CutOut on the controlled testbed, varying the size $c$ of erased patches in augmented images. For example, CutOut-0.2 represents that the width of an erased patch is 20% of that of each image, making their relative area 4%. Note that the original CutOut used as `gen` selects the patch width randomly in $(0.02, 0.33)$, making an average of 0.19 and thus aligning best with $c = 0.2$.

The figure shows that CutOut-$c$ performs better with smaller values of $c$, and it starts to decrease the AUC of the unsupervised baseline when $c \geq 0.6$. This is because CutOut-$c$ with small $c$ achieves the best alignment with `gen`, which removes small patches of average size $\approx 0.2$. Although the functional type of `aug` is the same as `gen`, the value of $c$ determines whether $f_{\text{aug}}$ succeeds or not.

---

[2]See *Everything is Obvious: *Once You Know the Answer.* Duncan J. Watts. Crown Business, 2011.

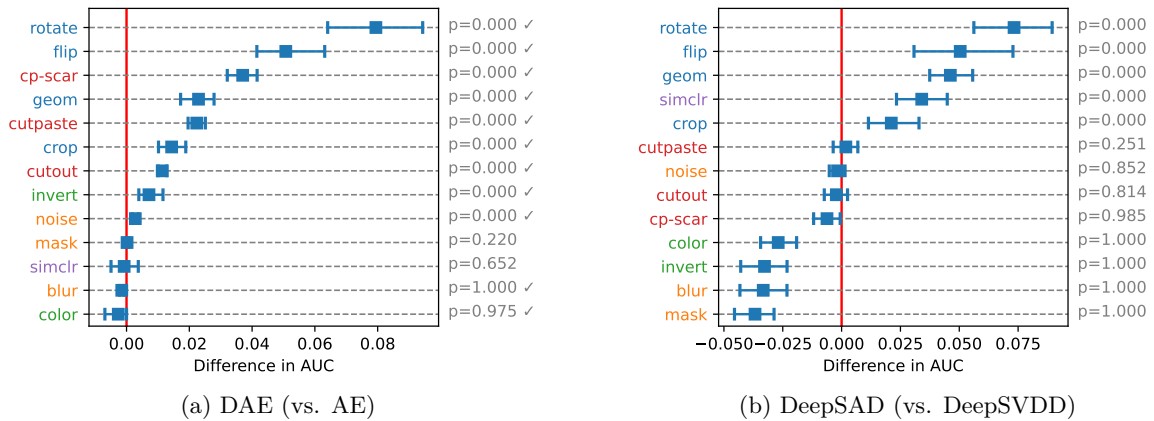

(a) DAE (vs. AE)  (b) DeepSAD (vs. DeepSVDD)

Figure 5: (best in color) Relative AUC on the in-the-wild testbed in which anomalies are different semantic classes. Color represents the category of augmentation. Geometric augmentations (in blue) perform best in both cases while others make insignificant changes or impair the baselines. See Obs. 4.

**Observation 2.** *Finding 1 is consistent with different hyperparameters of a detector model $f$.*

Our study aims to make observations that are generalizable across different settings of detector models and hyperparameters. To that end, we run experiments on in-the-wild testbed with 16 hyperparameter settings of DAE, making a total of 5,760 tasks (4 datasets × 90 class pairs × 16 settings): the number of epochs in $\{64, 128\}$, the number of features in $\{128, 256\}$, weight decay in $\{10^{-5}, 5 \times 10^{-5}\}$, and hidden dimension size in $\{64, 128\}$. Note that the hyperparameters of AP and DeepSAD are directly taken from previous works.

Fig. 4b presents the results, which are directly comparable to Fig. 5a. Both figures show almost identical orders of `aug` functions only with slight differences in numbers. This suggests that although the absolute AUC values are affected by model hyperparameters, the relative AUC with respect to the vanilla AE is stable since we use the same hyperparameter setting for DAE and AE. A notable difference is that the confidence intervals are smaller in Fig. 4b than in Fig. 5a due to the increased number of trials from 90 to 5,760.

### 4.3 Additional Observations: Error Bias and Semantic Anomalies

Based on our main finding, we introduce additional observations toward a better understanding of the effect of self-supervision on SSAD: error bias (in Obs. 3) and the alignment on semantic anomalies (in Obs. 4).

**Observation 3.** *Given a dataset containing multiple types of anomalies, self-supervision with `aug` creates a bias in the error distribution of $f_{\mathsf{aug}}$ compared with that of the vanilla $f$.*

Fig. 3 compares DAE and DeepSAD with their no-SSL baselines, AE and DeepSVDD, respectively, on multiple types of anomalies on CIFAR-10. In Fig. 3a, $f_{\mathsf{aug}}$ decreases the accuracy of $f$ on Automobile and Truck, which are semantic classes that include ground (or dirt) in images and thus can be easily separated from Airplane by unsupervised learning. The self-supervision with Rotate forces $f_{\mathsf{aug}}$ to detect other semantic classes including sky as anomalies, such as Bird, by feeding rotated airplanes as pseudo anomalies during the training. Such a bias is observed similarly in Fig. 3b with DAE, while the amount of bias is smaller than in Fig. 3a. This result shows that the "bias" phenomenon existing in semi-supervised learning (Ye et al., 2021) is observed also in SSL, and emphasizes the importance of selecting a proper `aug` function especially when there exist anomalies of multiple semantic types.

**Observation 4.** *Geometric augmentations work best when anomalies are different semantic classes, consistent with similar observations in previous works (Golan & El-Yaniv, 2018; Li et al., 2021).*

Fig. 5 shows the results on 360 in-the-wild tasks across four datasets and 90 class pairs each, whose anomalies represent different semantic classes in the datasets. The alignment between `aug` and `gen` is not known a priori unlike in the controlled testbed. The geometric `aug` functions such as Rotate and Flip work best with both DAE and DeepSAD, showing their effectiveness in detecting semantic class anomalies, consistent with the observations in previous works (Golan & El-Yaniv, 2018; Li et al., 2021). One plausible explanation is that many classes in those datasets, such as dogs and cats in CIFAR-10, are sensitive to geometric changes such as

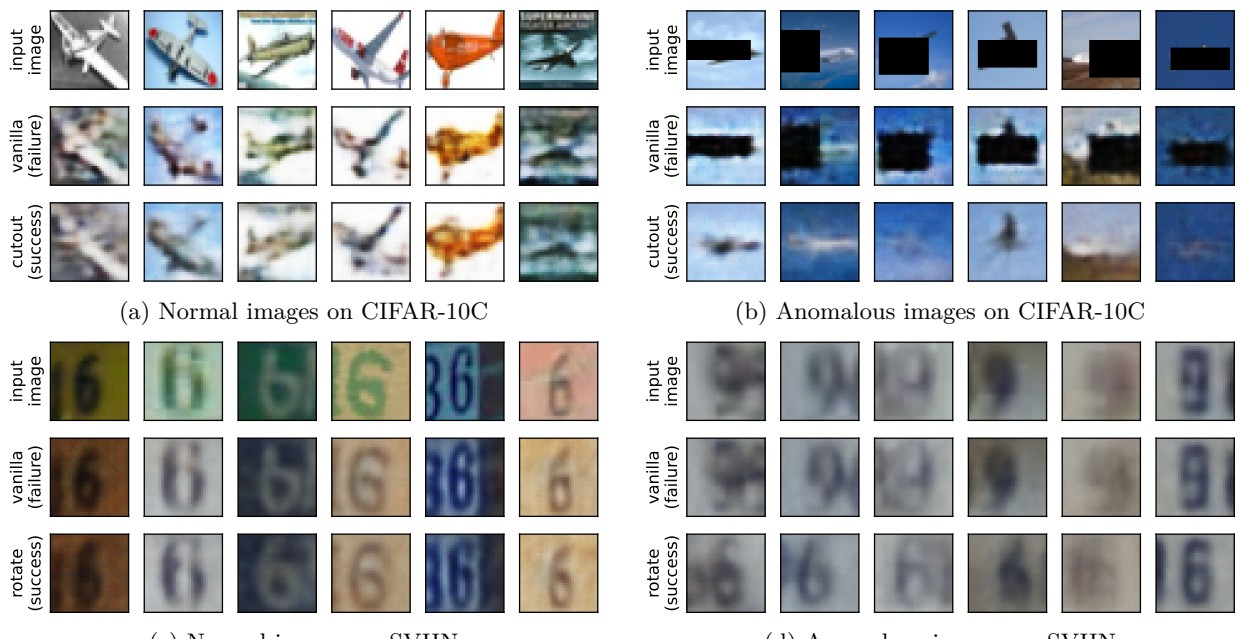

(a) Normal images on CIFAR-10C        (b) Anomalous images on CIFAR-10C

(c) Normal images on SVHN        (d) Anomalous images on SVHN

Figure 6: Images from CIFAR-10C and SVHN, where the three rows represent original images and those reconstructed by AE $f$ and DAE $f_{\mathsf{aug}}$, respectively. (a, b) aug:=CutOut and gen:=CutOut. (c, d) aug:=Rotate, and the digits 6 and 9 are normal and anomalous classes, respectively. The **success** and **failure** represent whether the images are assigned accurately to their true classes (normal vs. anomaly). DAE preserves the original normal images (on the left) while applying $\mathsf{aug}^{-1}$ to anomalies (on the right), making them resemble normal ones with high reconstruction errors. As a result, DAE $f_{\mathsf{aug}}$ produces high AUC of (a, b) 0.986 and (c, d) 0.903, while those of AE $f$ are low as (a, b) 0.865 and (c, d) 0.549, respectively. See Obs. 5.

rotation. Thus, aug creates plausible samples outside the distribution of normal data, giving $f_{\mathsf{aug}}$ an ability to differentiate anomalies that may look like augmented (i.e. rotated) normal images.

## 5 How Augmentation Works

Based on our findings and observations in Sec. 4, we perform case studies and detailed analyses to study how augmentation helps anomaly detection. We adopt DAE as the main model of analysis due to the following reasons. First, DAE learns to reconstruct the original sample $\mathbf{x}$ from its corrupted (i.e., augmented) version $\mathsf{aug}(\mathbf{x})$, which helps with the interpretation of anomalies. Second, DAE is simply its unsupervised counterpart AE when $\mathsf{aug} = \text{Identity}$, which helps study the effect of other augmentations on DAE.

### 5.1 Case Studies on CutOut and Rotate Functions

We visually inspect individual samples to observe what images DAE reconstructs for different inputs. Fig. 6 shows the results on CIFAR-10C and SVHN with different aug functions.

**Observation 5.** *Given a normal sample* $\mathbf{x}$*, DAE* $f_{\mathsf{aug}}$ *approximates the identity function* $f(\mathbf{x}) \approx \mathbf{x}$*. Given an anomaly* $\mathsf{gen}(\mathbf{x})$*,* $f_{\mathsf{aug}}$ *approximates the inverse* $\mathsf{aug}^{-1}$ *if* gen *and* aug *are aligned, i.e.,* $f_{\mathsf{aug}}(\mathsf{gen}(\mathbf{x})) \approx \mathbf{x}$*.*

We observe from Fig. 6 that both DAE and vanilla AE produce low reconstruction errors for normal data, but AE fails to predict them as normal since the errors are low also for anomalies. Given anomalies, DAE recovers their counterfactual images by applying $\mathsf{aug}^{-1}$, increasing their reconstruction errors to be higher than those from the normal images and achieving higher AUC than that of AE. It is noteworthy that the task to detect digits 9 as anomalies from digits 6 is *naturally aligned* with the Rotate augmentation function because, in effect, DAE learns the images of rotated 6 to be anomalies during training. We show in Appendix D that similar observations are derived from other pairs of normal classes and anomalies.

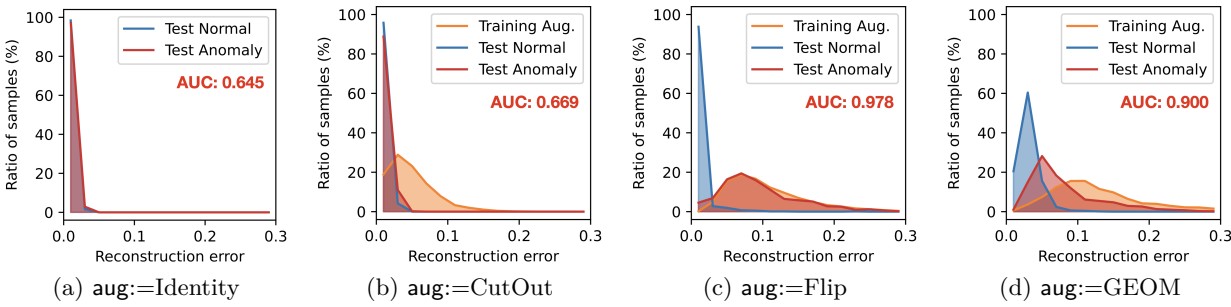

Figure 7: (best in color) Reconstruction error distributions on CIFAR-10C with Automobile as the normal class and gen:=Flip. The distributions gradually shift to the right as aug changes the input images more and more: (a) Identity, (b) CutOut (local), (c) Flip (global), and (d) GEOM ("cocktail" augmentation). The distributions of augmented samples and anomalies are matched the most in (c) when aug = gen, which also achieves the smallest MMD: (a) 0.063, (b) 0.277, (c) 0.031, and (d) 0.399. See Obs. 6.

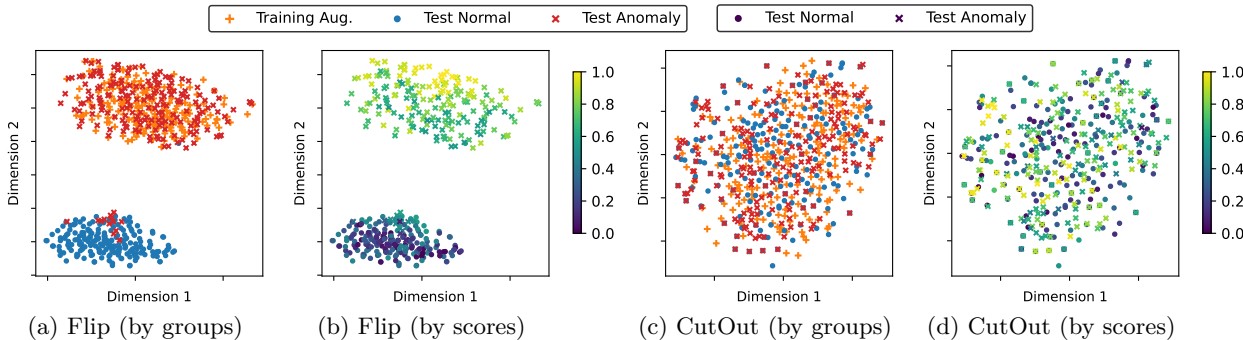

Figure 8: (best in color) $t$-SNE visualization of data embeddings on CIFAR-10C under perfect alignment: (a, b) aug = gen = Flip and (c, d) aug = gen = CutOut. The colors represent either (a, c) data categories or (b, d) anomaly scores. DAE achieves high AUC of 0.978 and 0.973 in both cases, respectively, despite the distributions of embeddings being different geometrically. See Obs. 7.

## 5.2 Error Histograms with Varying Degrees of Modification

Next, we study the effect of augmentation on the distribution of reconstruction errors on normal data and anomalies. Fig. 7 shows the error histograms on CIFAR-10C with different aug functions.

**Observation 6.** *The overall reconstruction errors are higher in DAE $f_{aug}$ than in vanilla AE $f$, and increase with the degree of modification that aug employs.*

We compare DAE with three different aug functions and AE in Fig. 7; recall that $f_{aug}$ with aug = Identity is equivalent to AE. In general, DAE shows higher errors than those of AE, leading to more right-shifted error distributions. This is because the training process of DAE involves a denoising operation that increases the reconstruction errors for augmented data by mapping them to normal ones. The improved AUC of DAE is the result of increasing reconstruction errors for anomalies than those for normal data.

In Fig. 7, the amount of change incurred by aug increases from Fig. 7a to 7d. The error distributions are shifted to the right accordingly, as augmented data become more different from the normal ones. Notably, the distribution for anomalies is more right-shifted in Fig. 7c than in Fig. 7d, resulting in higher detection AUC of 0.978, due to the better alignment between aug and gen. This showcases that the amount of alignment between aug and gen is the main factor that determines the errors (i.e., anomaly scores) of anomalies, while the general distributions are affected by the amount of modification made by aug. We show in Appendix E that our observation is consistent in different gen functions for various tasks.

### 5.3 Embedding Visualization: Clusters and Separability

We visualize data embeddings generated by DAE to understand how the embedding distributions of augmented training samples, test normal, and test anomalous samples affect the detection performance.

**Observation 7.** *Data embeddings generated by DAE $f_{\mathsf{aug}}$ for normal data and anomalies are separated when* $\mathsf{aug}$ *makes global changes in images, whereas mixed when* $\mathsf{aug}$ *makes local changes.*

Fig. 8 illustrates data embeddings on the controlled testbed when $\mathsf{aug}$ and $\mathsf{gen}$ are the same. The first two and the last two figures exhibit different scenarios. In Fig.s 8a and 8b, there exist separate clusters: one for normal data, and another for augmented data and anomalies. In Fig.s 8c and 8d, all data compose a single cluster without a separation between the different groups, even with the high AUC of 0.973. The difference between the two scenarios is mainly driven by the characteristic of the $\mathsf{aug}$ function, especially the amount of modification made by $\mathsf{aug}$: Flip makes global changes, while CutOut affects only a part of each image.

Fig. 9 supports Obs. 7 through embeddings with different patch sizes of CutOut as in Fig. 4a. Augmented data start to create dense clusters as the patch size $c$ increases, and they are completely separated from the normal data and anomalies when $c = 0.9$ in Fig. 9d. One notable difference from Fig. 8 is that Fig.s 9c and 9d represent failures, as the training augmented data are separated from both training normal data and test anomalies, while Fig. 8a represents a success due to the alignment between $\mathsf{aug}$ and $\mathsf{gen}$.

## 6 Discussion

**Summary of Findings and Take-aways.** In this study, we took a deeper look at the role of augmentation in self-supervised anomaly detection (SSAD) through extensive carefully-designed experiments. Our findings and observations are summarized as follows:

- Success and failure: The alignment between augmentation $\mathsf{aug}$ and anomaly-generating mechanism $\mathsf{gen}$ determines the success of SSAD (Finding 1 and Obs. 2). The degree of alignment is determined not only by the functional type of $\mathsf{aug}$, but also by its continuous hyperparameters (Obs. 1).
- Empirical performance and bias: Geometric augmentation works best on datasets in which anomalies are different semantic classes (Obs. 4). When there exist multiple types of anomalies, self-supervision leads to a bias in the error distribution as observed also in semi-supervised learning (Obs. 3).
- Case studies: Given an anomaly, DAE approximates the inverse $\mathsf{aug}^{-1}$ of augmentation if $\mathsf{gen}$ and $\mathsf{aug}$ are aligned (Obs. 5), and it increases overall reconstruction errors (Obs. 6). The distribution of data embeddings can be the key to estimate the degree of alignment between $\mathsf{aug}$ and $\mathsf{gen}$ (Obs. 7).

Such findings clearly demonstrate that SSL for AD emerges as a data- or task-specific solution, rather than a cure-all panacea, effectively rendering it a critical hyperparameter. What makes this a nontrivial problem is that finding a good augmentation function is challenging in fully unsupervised settings if there is no prior knowledge of unseen anomalies. Our findings pave a path toward possible solutions as discussed below.

**Future Research Directions.** We suggest *transductive learning* as a future direction of SSAD, which is to assume unlabeled test data are given at training time. Vapnik (2006) has been an advocate of transductive learning, according to whom, one should not solve a more general and harder intermediate problem and then try to induce to a specific one, but rather solve the specific problem at hand directly. In our setting, the use of unlabeled test data transductively, containing the anomalies to be identified, opens a possibility to tune the augmentation without accessing any labels. At the same time, it creates a fundamental difference from existing approaches that *imagine* how the actual anomalies would look like or otherwise haphazardly choose an augmentation function, which may not well align with what is to be detected.

In transductive SSAD setting, the distribution of data embeddings can offer the grounds for an unsupervised measure of semantic alignment between $\mathsf{aug}$ and $\mathsf{gen}$. Fig.s 8 and 9 show that the embeddings of augmented data are overlapped with those of anomalies under the perfect alignment, while they are separated without alignment. Transductive learning allows us to approximate the distance between $\mathsf{aug}$ and $\mathsf{gen}$ in the form of $\mathrm{dist}(\mathcal{D}_{\mathrm{trn}} \cup \mathcal{D}_{\mathrm{aug}}, \mathcal{D}_{\mathrm{test}})$, where $\mathrm{dist}(\cdot, \cdot)$ is a set distance function, and $\mathcal{D}_{\mathrm{trn}}$, $\mathcal{D}_{\mathrm{aug}}$, and $\mathcal{D}_{\mathrm{test}}$ are sets of training normal, training augmented, and unlabeled test data, respectively. Since $\mathcal{D}_{\mathrm{test}}$ contains both normal data and (unlabeled) anomalies, its distance to $\mathcal{D}_{\mathrm{trn}} \cup \mathcal{D}_{\mathrm{aug}}$ is expected to be small when they align well.

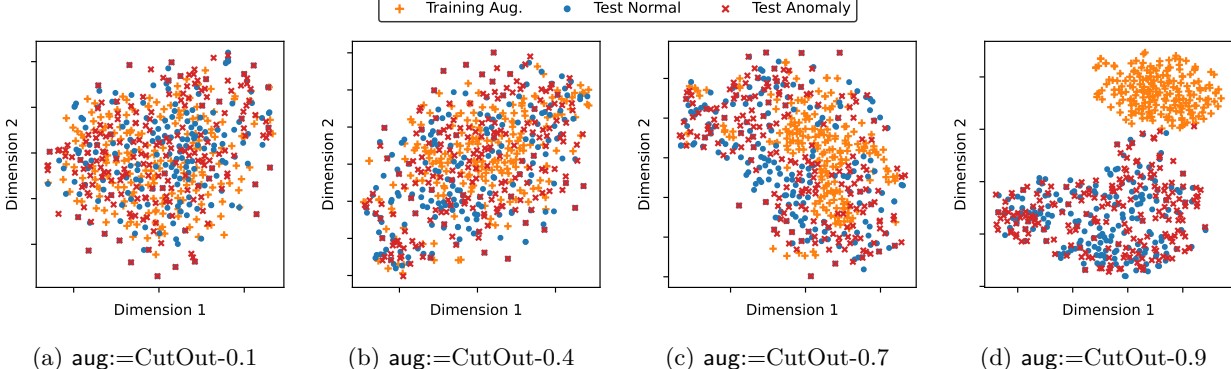

(a) aug:=CutOut-0.1   (b) aug:=CutOut-0.4   (c) aug:=CutOut-0.7   (d) aug:=CutOut-0.9

Figure 9: $t$-SNE visualization of data embeddings on CIFAR-10C where aug:=CutOut-$c$ and gen:=CutOut. The value of $c$ represents the patch size of CutOut as in Fig. 4a. The augmented samples and anomalies are matched well in (a), but start to be separated as the alignment between aug and gen becomes weaker from (b) to (d). The result is consistent with Fig. 4a, and supports Obs. 7.

Towards this end, recent work has designed an unsupervised validation loss for model selection for SSAD based on a quantitative measure of alignment in the embedding space (Yoo et al., 2023b). They have later improved this loss into a differentiable form toward augmentation hyperparameter tuning in an end-to-end framework for image SSAD (Yoo et al., 2023a). Future research could continue to design new unsupervised losses as well as differentiable augmentation functions amenable for end-to-end tuning. When equipped with those, one can then systematically apply modern HPO techniques (Bischl et al., 2023) to SSAD.

## 7    Conclusion

In this work, we studied the role of self-supervised learning (SSL) in unsupervised anomaly detection (AD). Through extensive experiments on 3 SSL-based detectors across 420 AD tasks, we showed that the alignment between data augmentation and anomaly-generating mechanism plays an essential role in the success of SSL; and importantly, SSL can even hurt detection performance under poor alignment. Our study is motivated (and our findings are also supported) by partial evidences or weaknesses reported in the growing body of SSAD literature, and serves as the first systematic meta-analysis to provide comprehensive evidence on the success or failure of SSL for AD. We expect that our work will trigger further research on better understanding this growing area, in addition to new SSAD solutions that can aptly tackle the challenging problem of tuning the data augmentation hyperparameters for unsupervised settings in a principled fashion.

## Acknowledgements

This work is partially sponsored by PwC Risk and Regulatory Services Innovation Center at Carnegie Mellon University. Any conclusions expressed in this material are those of the author and do not necessarily reflect the views, expressed or implied, of the funding parties.

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

## A    Detailed Information on Detector Models

In our experiments, we use the model structures used in previous works. For AE and DAE, we adopt the structure used in (Golan & El-Yaniv, 2018). The encoder and decoder networks consist of four encoder and decoder blocks, respectively. Each encoder block has a convolution layer of $3 \times 3$ kernels, batch normalization, and a ReLU activation function. A decoder block is similar to the encoder block, except that the convolution operator is replaced with the transposed convolution of the same kernel size. The number of epochs and the size of hidden features are both set to 256, and the number of convolution features is $(64, 128, 256, 512)$ for the four layers of the encoder block, respectively.

For AP (Golan & El-Yaniv, 2018) and DeepSAD (Ruff et al., 2020b), we use their official implementations in experiments.[3][4] The structure of AP is based on Wide Residual Network (Zagoruyko & Komodakis, 2016), and DeepSAD is based on a LeNet-type convolutional neural network.

## B    Detailed Information on Augmentation Functions

We provide detailed information on augmentation functions that we study in this work. We use the official PyTorch implementations of augmentation functions and their default hyperparameters when available.

**Geometric augmentations** modify images with geometric functions:

- Rotate (random rotation) makes a random rotation of an image with a degree in $\{0, 90, 180, 270\}$.
- Crop (random cropping) randomly selects a small patch from an image whose relative size is between 0.08 and 1.0, resizes it to the original size, and uses it instead of the given original image.
- Flip (vertical flipping) vertically flips an image.
- GEOM (Golan & El-Yaniv, 2018) applies three types of augmentations at the same time (and in this order): Rotate, Flip, and Translate, where Translate denotes a random horizontal or vertical translation by 8 pixels.

**Local augmentations** change only a small subset of image pixels without affecting the rest.

- CutOut (random erasing) (Devries & Taylor, 2017; Zhong et al., 2017) erases a small patch from an image, replacing the pixel values as zero (i.e., black pixels). The patch size is chosen randomly from $(0.02, 0.33)$.
- CutPaste (Li et al., 2021) copies a small patch and pastes it into another location in the same image. The difference from CutOut is that CutPaste has no black pixels in resulting images, making them more plausible. The patch size is chosen from $(0.02, 0.15)$.
- CutPaste-scar (Li et al., 2021) is a variant of CutPaste, which augments thin scar-like patches instead of rectangular ones. The patch width and height are chosen from $(10, 25)$ and $(2, 16)$, respectively, in pixels. The selected patches are rotated randomly with a degree in $(-45, 45)$ before they are pasted.

**Elementwise augmentations** make a change in the value of each image pixel individually (or locally).

- Noise (addition of Gaussian noise) (Vincent et al., 2010) is a traditional augmentation function used for denoising autoencoders. It adds a Gaussian noise with a standard deviation of 0.1 to each pixel.
- Mask (Bernoulli masking) (Vincent et al., 2010) conducts a random trial to each pixel whether to change the value to zero or not. The masking probability is 0.2.
- Blur (Gaussian blurring) (Chen et al., 2020) smoothens an image by applying a Gaussian filter whose kernel size is 0.1 of the image. The $\sigma$ of the filter is chosen randomly from $(0.1, 2.0)$ as done in the SimCLR function.

**Color augmentations** change the color information of an image without changing actual objects.

- Color (Color jittering) (Chen et al., 2020) creates random changes in image color with brightness, contrast, saturation, and hue. The amount of changes is the same as in the SimCLR function.

---

[3]https://github.com/izikgo/AnomalyDetectionTransformations
[4]https://github.com/lukasruff/Deep-SAD-PyTorch

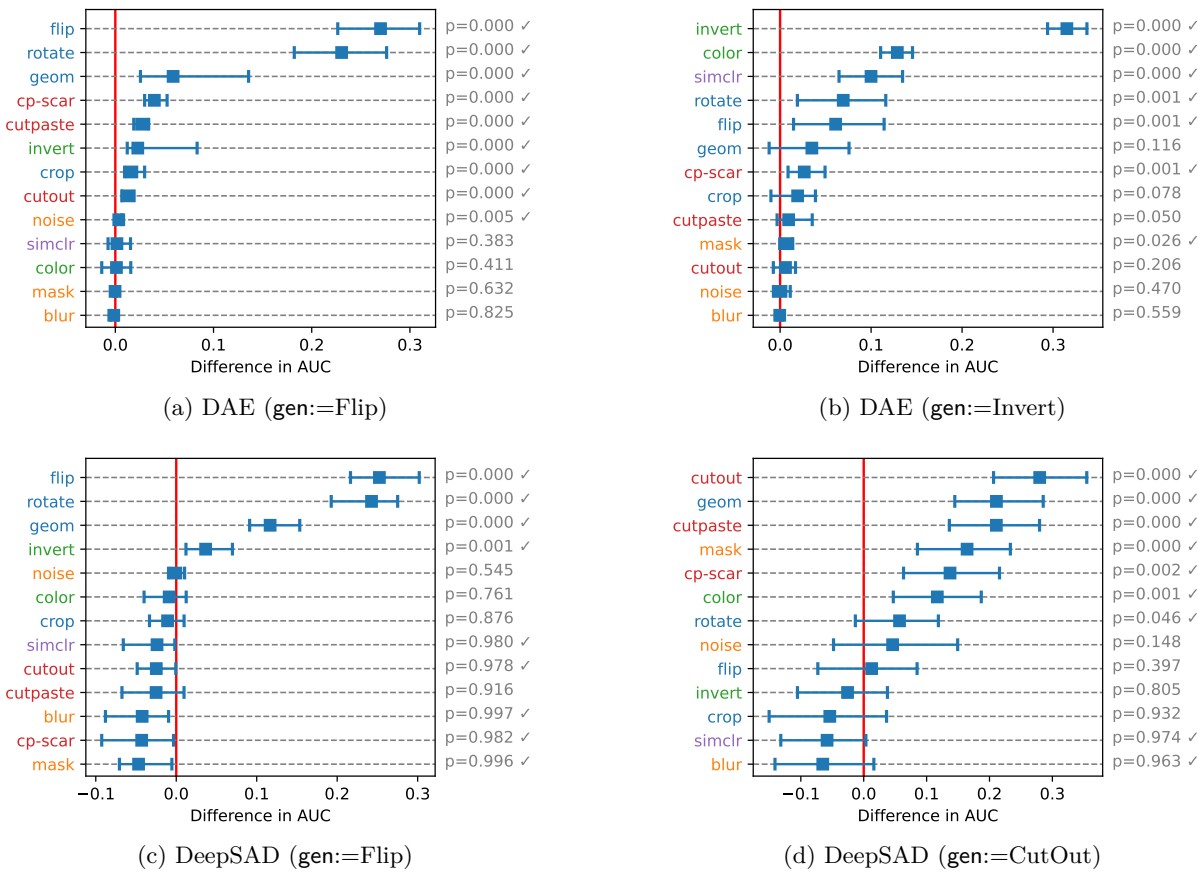

Figure 10: (best in color) Relative AUC of (a, b) DAE and (c, d) DeepSAD with respect to the no-SSL baselines on the controlled testbed with anomaly-generating functions (a, c) **gen**:=Flip, (b) **gen**:=Invert, and (d) **gen**:=CutOut. Color in augmentation names denotes their categories. Augmentation functions aligned with the anomaly-generating functions perform well, supporting Finding 1.

- Invert (Color inversion) inverts the color information of an image. In the actual implementation, it returns $1 - x$ for each pixel $x \in [0, 1]$.

**Mixed augmentations** combine augmentations of multiple categories, making unified changes.

- SimCLR (Chen et al., 2020) has been used widely in the literature as a general augmentation function (Tack et al., 2020). It conducts the following at the same time (and in the given order): cropping, horizontal flipping, color jittering, gray scaling, and Gaussian blurring.

## C    Full Results on Individual Datasets for Main Finding

We present more results of relative AUC on DAE and DeepSAD, compared with their no-SSL baselines. Our additional experiments support Finding 1 and Observation 4, which are presented informally as follows:

- **Finding 1:** Self-supervised detection performs better with a better alignment between **aug** and **gen** functions, and fails (i.e., performs worse than the unsupervised baseline) under poor alignment.
- **Obs. 4:** Geometric **aug** works best with synthetic class anomalies (i.e., in-the-wild testbed).

Fig. 10 shows the relative AUC on the controlled testbed across two datasets CIFAR-10C and SVHN-C and ten classes each. The missing combinations, DAE with **gen**:=CutOut and DeepSAD with **gen**:=Invert, are given in the main paper. All four cases in the figure with different models and anomaly types support our main finding, showing the generalizability of our work into various combinations of **aug** and **gen**.

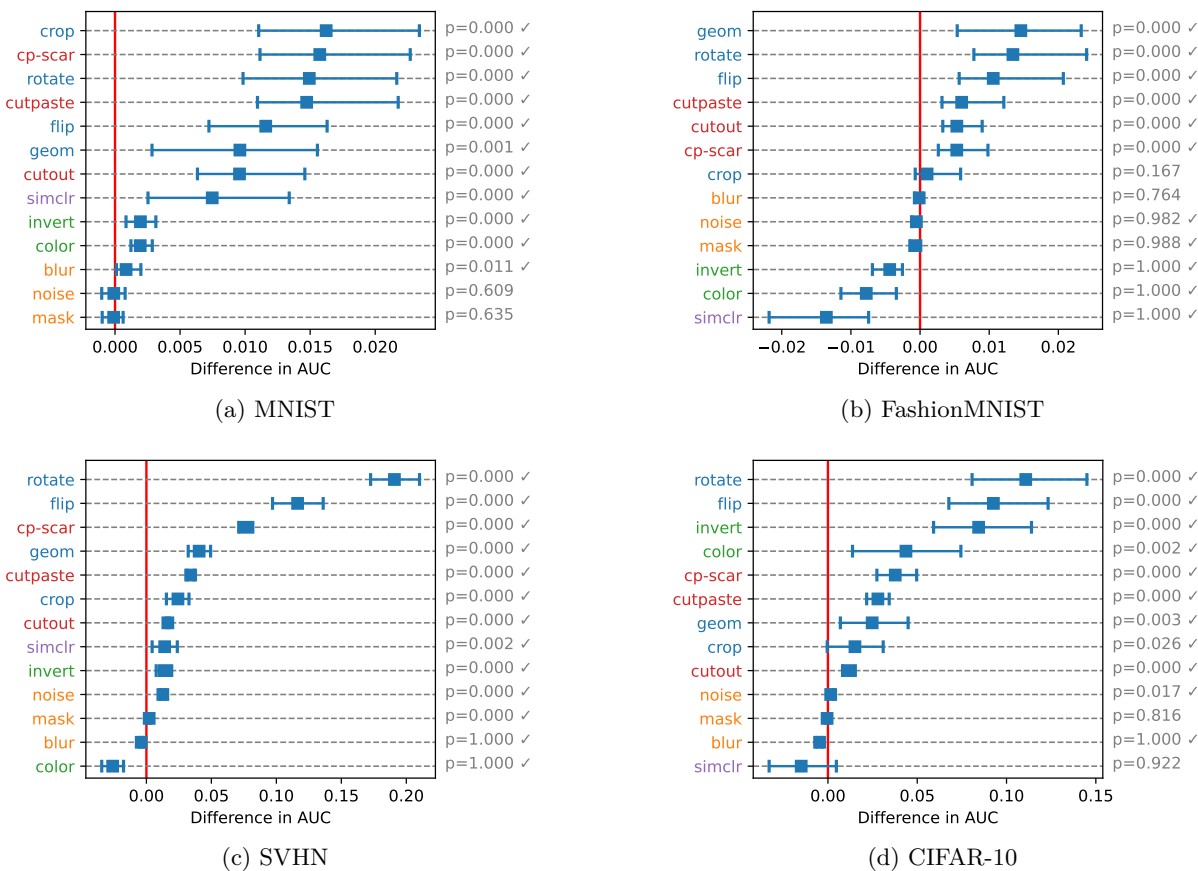

Figure 11: (best in color) Relative AUC of DAE with respect to its no-SSL baseline, AE, on the in-the-wild testbed in which anomalies are different semantic classes: (a) MNIST, (b) FashionMNIST, (c) SVHN, and (d) CIFAR-10. Color in the augmentation names denotes their categories. The geometric augmentations (in blue) perform best in all datasets, improving the baselines, while others make insignificant changes or impair the baselines. The patterns are consistent with Fig. 5a in the main paper. See Obs. 4.

Fig.s 11 and 12 show the relative AUC on the in-the-wild testbed for four individual datasets. That is, Fig.s 5a and 5b are statistical summaries of Fig. 11 and 12, respectively, over the four datasets. The results show that every dataset in the testbed exhibits Obs. 4, although the characteristics of datasets are different from each other, supporting the generality of our findings toward different datasets.

# D    Case Studies on Other Augmentation and Anomaly Functions

We conduct additional case studies to support Obs. 5 on different types of `gen` functions. Our observation is presented informally as follows:

- **Obs. 5:** DAE applies $\mathsf{aug}^{-1}(\cdot)$ to anomalies, while making no changes on normal data.

**Controlled Testbed.** Fig. 13 shows images in CIFAR-10C when (top) `gen`:=Flip and (bottom) `gen`:=Invert. DAE applies the inverse augmentation $\mathsf{aug}^{-1}$ to the anomalies, reconstructing normal-like ones; in Fig. 13b, the given images are rotated in the reconstructed ones when `aug`:=Flip, while in Fig. 13d, the color of given anomalous images is inverted back in the reconstructed ones when `aug`:=Invert. In contrast, the vanilla AE reconstructs both normal and anomalous images close to the input images, as stated in Obs. 5.

**In-the-Wild Testbed.** Fig. 14 shows the images of SVHN, in which anomalies are images associated with different digits. We study four `aug` functions (from top to bottom): Flip, CutOut, CutPaste, and SimCLR.

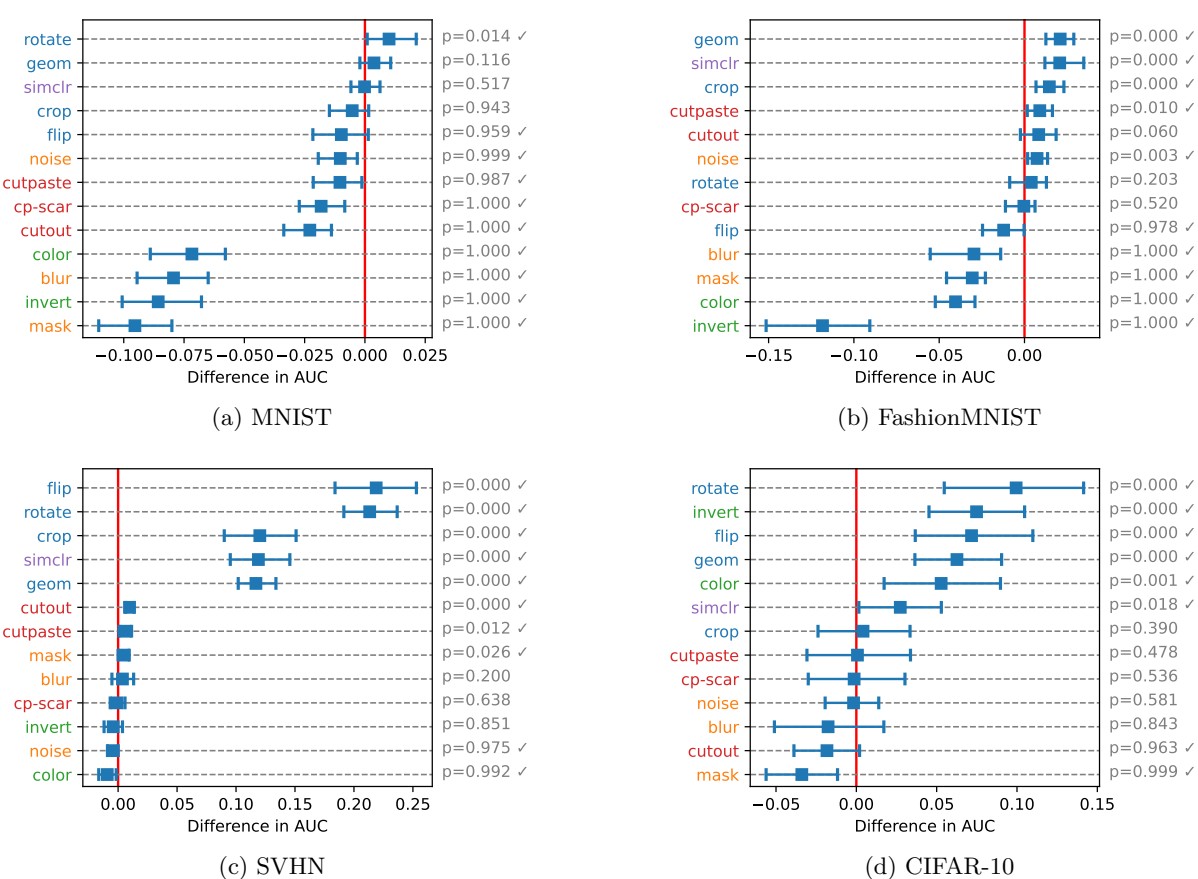

Figure 12: (best in color) Relative AUC of DeepSAD with respect to its no-SSL baseline, DeepSVDD, on the in-the-wild testbed in which anomalies are different semantic classes: (a) MNIST, (b) FashionMNIST, (c) SVHN, and (d) CIFAR-10. Color in the augmentation names represents their categories. The geometric augmentations (in blue) perform best in all datasets, improving the baselines, while others make insignificant changes or impair the baselines. The patterns are consistent with Fig. 5b. See Obs. 4.

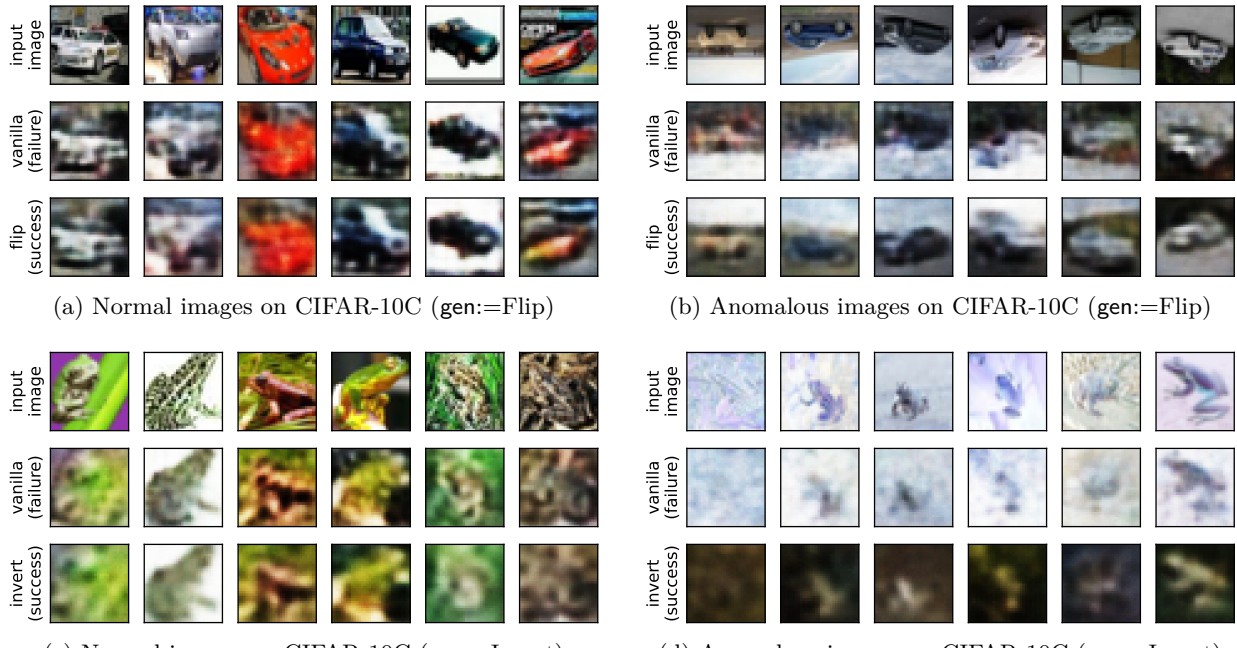

(a) Normal images on CIFAR-10C (gen:=Flip)

(b) Anomalous images on CIFAR-10C (gen:=Flip)

(c) Normal images on CIFAR-10C (gen:=Invert)

(d) Anomalous images on CIFAR-10C (gen:=Invert)

Figure 13: Images from CIFAR-10C, where the three rows represent original images and those reconstructed by the vanilla AE and DAE, respectively: (a, b) aug = gen = Flip with Automobile as the normal class, and (c, d) aug = gen = Invert with Frog as the normal class. The **success** and **failure** represent whether the images are assigned accurately to their true classes or not (normal vs. anomaly). DAE preserves the original images (on the left), while applying $aug^{-1}$ to anomalies (on the right), making them resemble normal ones with high reconstruction errors. This allows DAE to achieve higher AUC than AE, supporting Obs. 5.

Task 1: 4 vs. 7. When aug:=Rotate and the task is 4 (normal) vs. 7 (anomalous), DAE successfully generates 4-like images from 7 by applying the inverse of Rotate, which is also a rotation operation but with a different degree. This is because the digit 7 can be considered or somewhat resembles rotated 4 as shown in Fig. 14b, as in the case of 6 vs. 9 in Fig.s 6c and 6d.

Task 2: 0 vs. 3. We study another task of 0 (normal) vs. 3 (anomalous) with the remaining aug functions. CutOut and CutPaste generally work well, since the scarred images of 0 can look like 3 by chance. CutOut works best with anomalous images of a black background, since the inverse function of CutOut is to fill in the black erased patch; if a given image has a white background, it is difficult to find such a patch to revert by $aug^{-1}$. CutPaste does not require an anomalous image to have a background of a specific color, since it copies and pastes an existing patch instead of erasing image pixels.

The SimCLR augmentation in Fig.s 14g and 14h, unlike the other aug functions, changes the information of color and shape at the same time. This is because SimCLR is a "cocktail" augmentation that creates a large degree of modification by combining multiple augmentation functions. The degree of change is significant even with normal images in Fig. 14g, which makes it perform poorly in terms of detecting anomalies based on anomaly scores; both normal and anomalous images are reconstructed into similar forms.

# E    Error Histograms on Other Augmentation and Anomaly Functions

We present more detailed results on error histograms to support Obs. 6 on different tasks and different types of gen functions. We informally present our observation as follows:

- **Obs. 6:** Reconstruction errors are higher in DAE than in AE, increasing with the degree of change.

**Controlled Testbed.** Fig. 15 shows the error histograms on CIFAR-10C with two types of gen: CutOut and Invert. The figure supports Obs. 6 by presenting identical patterns as in Fig. 7 even with different gen

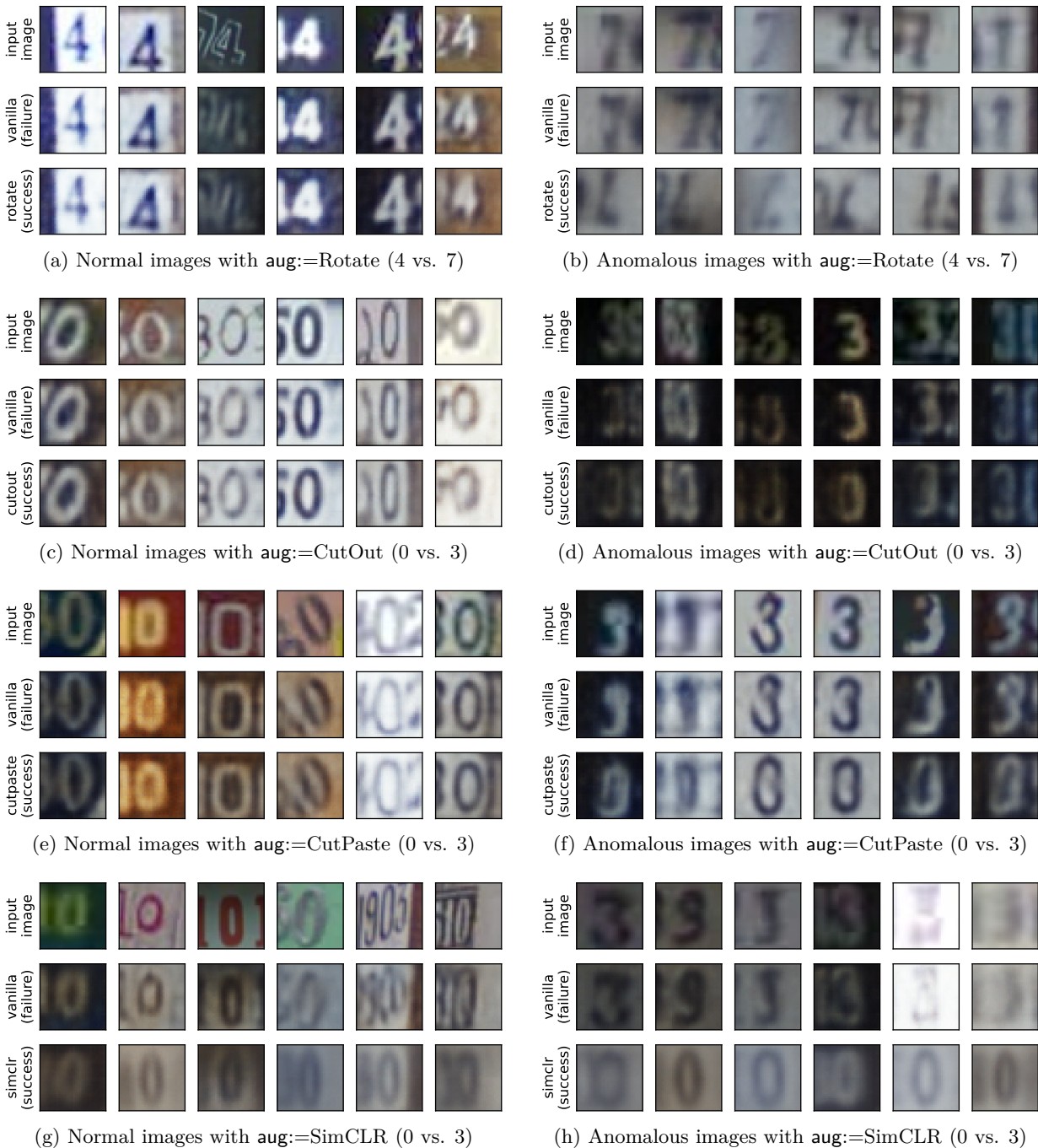

(a) Normal images with **aug**:=Rotate (4 vs. 7)    (b) Anomalous images with **aug**:=Rotate (4 vs. 7)

(c) Normal images with **aug**:=CutOut (0 vs. 3)    (d) Anomalous images with **aug**:=CutOut (0 vs. 3)

(e) Normal images with **aug**:=CutPaste (0 vs. 3)    (f) Anomalous images with **aug**:=CutPaste (0 vs. 3)

(g) Normal images with **aug**:=SimCLR (0 vs. 3)    (h) Anomalous images with **aug**:=SimCLR (0 vs. 3)

Figure 14: Images from SVHN, where the three rows represent original images and those reconstructed by AE and DAE, respectively. (a, b) Rotate works similarly as in Fig.s 6c and 6d, since the digit 7 resembles the rotated digit 4. Given the anomalous images of 7, the DAE recovers the 4-like images by rotation. (c, d, e, f) DAE with CutOut and CutPaste works well in the task 0 (normal) vs. 3 (anomalous), because the images of 0 with erased patches can mimic those of 3. (g, h) SimCLR changes not only the shapes of digits, but also the color information due to the high degree of augmentation. The results support Obs. 5.

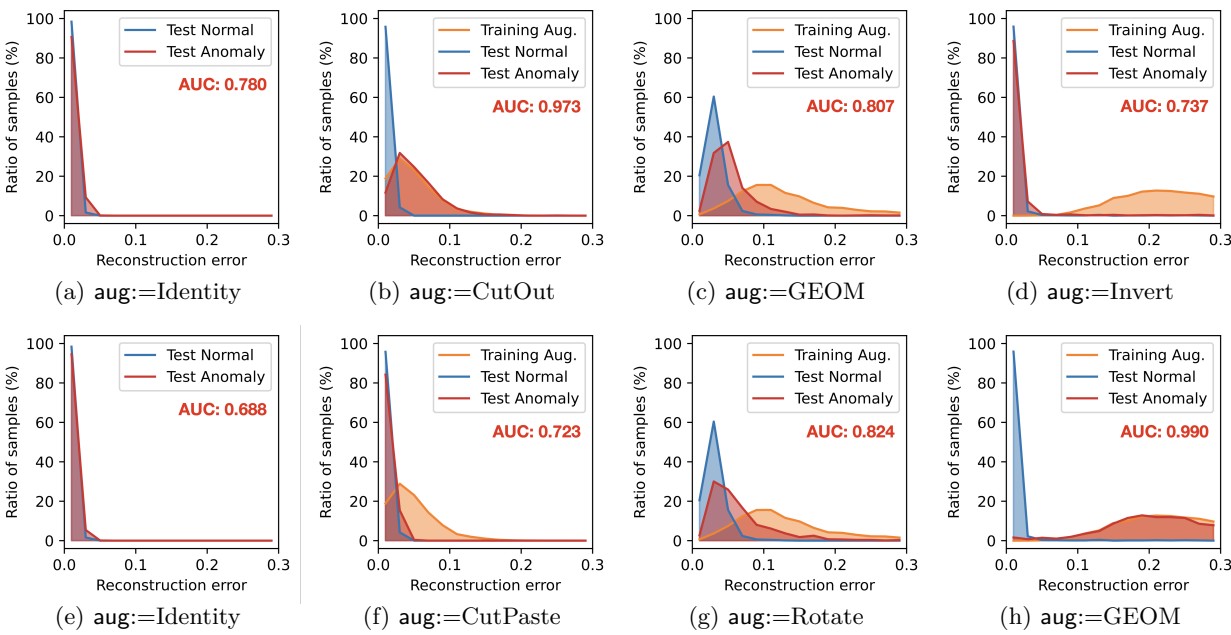

Figure 15: (best in color) Reconstruction errors on CIFAR-10C with Automobile as the normal class and gen:=CutOut (in the top row) and gen:=Invert (in the bottom row). The distributions gradually shift to the right as the augmentation aug changes the input images more and more: (a) Identity, (b) CutOut (local), (c) GEOM ("cocktail" augmentation), and (d) Invert (color-based), which inverts the values of all pixels. The distributions of augmented data and anomalies are matched the most in (b) and (h) when aug = gen, which is the case of perfect alignment, consistently with Fig. 7. The results support Obs. 6.

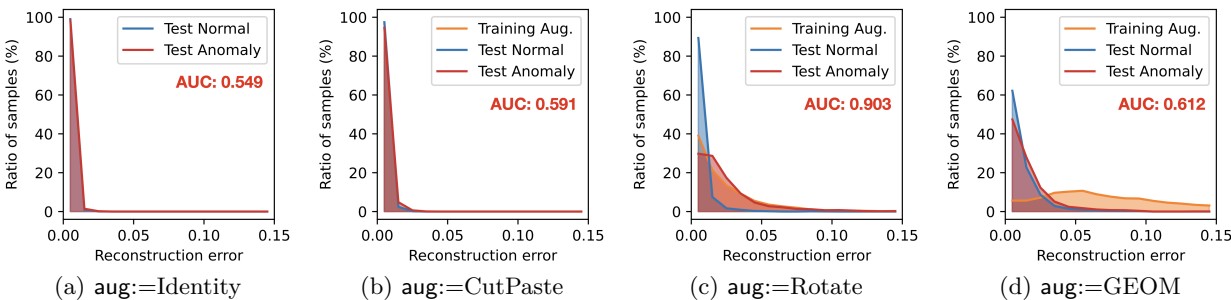

Figure 16: (best in color) Reconstruction errors on the SVHN data for 6 (normal) vs. 9 (anomalous). Similar observations are derived as in Fig. 7, since the gen function here can be thought of as approximate Rotate. Still, the error distribution of anomalies is not as clearly separated as in Fig.s 7 and 15, since the alignment between aug and gen is not perfect. The results support Obs. 6 on the in-the-wild testbed.

functions. A notable observation is that the distribution of augmented data is more right-shifted in Invert than in GEOM, even though GEOM is a "cocktail" approach that combines multiple augmentations. This is because Invert changes the value of every pixel simultaneously to invert the color of an image, which results in a dramatic change with respect to pixel values.

**In-the-Wild Testbed.** Fig. 16 shows the reconstruction errors on the SVHN dataset with different aug functions. The task is 6 (normal) vs. 9 (anomaly), where gen can be thought of as approximate rotation. Rotate works best among the four aug options thanks to the best alignment with gen. GEOM makes the most right-shifted distribution of augmented data, as in Fig. 7, due to its "cocktail" nature. The result on SVHN shows that Obs. 6 is valid not only for the controlled testbed, but also for the in-the-wild testbed.

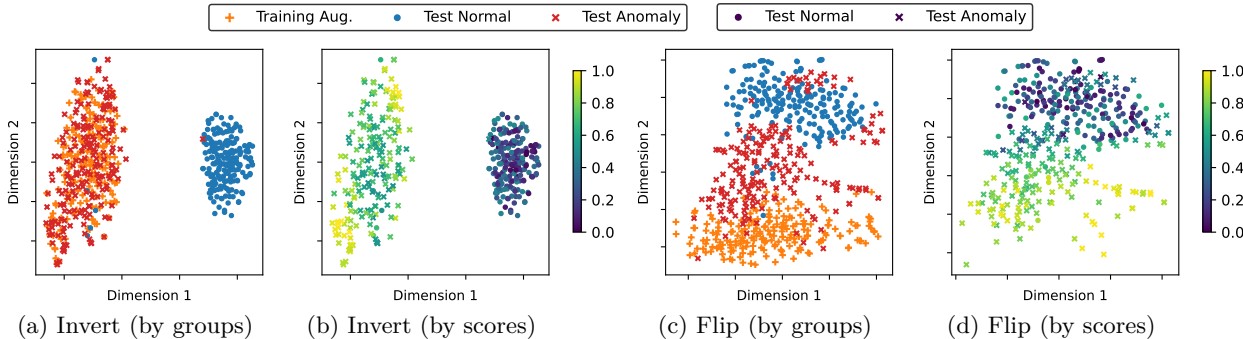

Figure 17: (best in color) $t$-SNE visualization of data embeddings on CIFAR-10C when **gen**:=Invert: (a, b) **aug**:=Invert and (c, d) **aug**:=Flip. The colors represent either (a, c) data categories or (b, d) anomaly scores. The first two and the last two cases show different patterns: (a, b) **aug**:=Invert creates two separate clusters as claimed in Obs. 7, achieving AUC of 0.990, thanks to the perfect alignment with **gen**. (c, d) **aug**:=Flip still achieves a high AUC of 0.889, although its alignment with **gen**:=Invert is unclear, since it puts anomalies in between normal and augmented data.

## F  Embedding Visualization on Other Augmentation and Anomaly Functions

We introduce more experimental results on the embedding visualization on both controlled and in-the-wild testbeds, supporting Obs. 7 with different combinations of **aug** and **gen** functions. We informally present our observation as follows:

- **Obs. 7:** Embeddings make separate clusters with global **aug**, and are mixed with local **aug**.

Fig. 17 shows the results on CIFAR-10C with **gen**:=Invert and two different **aug** functions. In Fig.s 17a and 17b, when **aug**:=Invert achieves the perfect alignment with **gen** and creates global changes in the image pixels through augmentation, the points make separate clusters supporting Obs. 7 and result in AUC of 0.990. In Fig.s 17c and 17d, when **aug**:=Flip exhibits imperfect alignment with **gen**, it still achieves high AUC of 0.889. This is because it succeeds in putting the anomalies in between normal and augmented samples in the embedding space. This allows $f$ to effectively detect the anomalies.

## G  Experiments with MMD

For the in-the-wild testbed, where **gen** is not given explicitly, we can utilize the maximum mean discrepancy (MMD) (Gretton et al., 2006) between augmented data and test anomalies to approximate the functional similarity. We create a set of augmented data by applying **aug** to the normal data. Then, we generate the embeddings of this set of data and test anomalies using the pretrained ResNet50 (He et al., 2016) as an encoder function, since pixel-level distance in raw images is not quite reflective of semantic differences in general. Lastly, we compute MMD between the embeddings of the two sets.

Given each task, let $\mathcal{D}_1$ and $\mathcal{D}_2$ be the sets of augmented data and test anomalies, respectively. First, we randomly sample $M$ instances from $\mathcal{D}_1$ and $\mathcal{D}_2$, respectively, and denote the results by $\mathcal{D}_1'$ and $\mathcal{D}_2'$. We set $M = 256$, which is large enough to make stable results over different random seeds. Then, MMD is computed between $\mathcal{D}_1'$ and $\mathcal{D}_2'$ as follows:

$$\text{MMD}(\mathcal{D}_1', \mathcal{D}_2') = \sum_{\mathbf{x}_1 \in \mathcal{D}_1'} \sum_{\mathbf{x}_1 \in \mathcal{D}_1'} k(\mathbf{x}_1, \mathbf{x}_1) + \sum_{\mathbf{x}_2 \in \mathcal{D}_2'} \sum_{\mathbf{x}_2 \in \mathcal{D}_2'} k(\mathbf{x}_2, \mathbf{x}_2) - 2 \sum_{\mathbf{x}_1 \in \mathcal{D}_1'} \sum_{\mathbf{x}_2 \in \mathcal{D}_2'} k(\mathbf{x}_1, \mathbf{x}_2), \tag{1}$$

where the kernel function $k(\cdot, \cdot)$ is defined on the outputs of a pretrained encoder network $\phi$:

$$k(\mathbf{x}_1, \mathbf{x}_2) = (\gamma \langle \phi(\mathbf{x}_1), \phi(\mathbf{x}_2) \rangle + c)^d. \tag{2}$$

Table 2: MMD between `aug` and `gen` on the in-the-wild testbed. The `aug` functions are ordered by the average distance. Geometric `aug` functions (in blue) exhibit the smallest distances in general, consistent with Fig. 5.

| Augment | MNI. | Fash. | CIF. | SVHN | Avg. |
|---------|------|-------|------|------|------|
| • Rotate | 1.831 | 1.855 | 0.267 | 0.186 | 1.035 |
| • CP-scar | 1.653 | 2.005 | 0.255 | 0.259 | 1.043 |
| • CutPaste | 1.697 | 2.100 | 0.258 | 0.203 | 1.065 |
| • GEOM | 1.754 | 1.774 | 0.423 | 0.446 | 1.099 |
| • Flip | 1.906 | 2.082 | 0.257 | 0.208 | 1.113 |
| • Color | 1.844 | 2.034 | 0.336 | 0.295 | 1.127 |
| • Invert | 2.221 | 2.116 | 0.257 | 0.220 | 1.204 |
| • Crop | 2.143 | 1.935 | 0.348 | 0.424 | 1.213 |
| • CutOut | 1.723 | 2.066 | 0.503 | 0.629 | 1.230 |
| • Blur | 2.182 | 2.372 | 0.303 | 0.221 | 1.270 |
| • SimCLR | 2.074 | 1.941 | 0.498 | 0.582 | 1.274 |
| • Mask | 2.136 | 2.824 | 0.585 | 0.776 | 1.580 |
| • Noise | 4.214 | 3.170 | 0.427 | 0.627 | 2.110 |
| Identity | 2.035 | 2.402 | 0.265 | 0.212 | 1.228 |

We adopt ResNet50 (He et al., 2016) pretrained on ImageNet as $\phi$, which works as a general encoder function that is independent of a detector model. The hyperparameters are set to the default values in the scikit-learn implementation: $\gamma = h^{-1}$, $c = 1$, and $d = 3$, where $h$ is the size of embeddings from $\phi$.[5]

Table 2 supports Obs. 4 with respect to the MMD between `aug` functions and the test anomalies. Geometric functions including Rotate, GEOM, and Flip show the smallest distances in general, representing that they are more aligned with the anomalies of different semantic classes than other `aug` functions are. One difference between Table 2 and Fig. 5 is that SimCLR, which shows large MMD on average, works better than the local `aug` functions (in red) in Fig. 5b. This is because the large flexibility of "cocktail" augmentation induced by the SimCLR is effective for DeepSAD, which learns a hypersphere that separates normal data from pseudo anomalies, while DAE aims to learn the exact mappings from pseudo anomalies to normal data.

---

[5]https://scikit-learn.org/stable/

