# OpenReview forum: "Data Augmentation is a Hyperparameter: Cherry-picked Self-Supervision for Unsupervised Anomaly Detection is Creating the Illusion of Success"
_TMLR — Accepted by TMLR_

### Review · Reviewer_aT5C · 2023-05-14

**Summary Of Contributions:**

This paper investigates the role of data augmentations in self-supervised learning-based (SSL) anomaly detection (AD) through extensive experiments and analysis. The major contributions of this research work, in my view, can be summarized as follows: 1. the comprehensive results of image-based SSL AD that can draw insights for understanding the effect of data augmentations; 2. bring clarity to the field of SSL AD using the alignment between pseudo and true anomalies. Overall this work presents the empirical exploration under a large lens.

**Audience:**

Yes

**Broader Impact Concerns:**

I think there is no related ethical concerns about this paper.

**Claims And Evidence:**

Yes

**Requested Changes:**

Overall I think this research work provides a comprehensive study for data augmentations in SSL AD, and I think the authors may consider adjusting the submission with the following points:
1. summarize all the observations for a clear presentation and provide the corresponding discussion if there is some relationship among all those empirical results.
2. discuss more about future suggestions after presenting the major insights of the comprehensive study, which may help to enhance the significance of this research study.

**Strengths And Weaknesses:**

Strengths:
1. The research problem is well motivated with a significant literature background, and empirically supported by those clear figures and clarifications.
2. The empirical study with 3 different types of SSAD models across 420 different AD tasks provides an extensive analysis of the role of data augmentations.
3. The details of the experimental setup and evaluations are clearly presented.

Weakness:
1. Although there are some interesting observations from those carefully designed experiments across so many experimental trials, would it be possible that we can conduct some theoretical analysis on the role of data augmentations to explain the phenomenon and guide future research in this direction? or provide some discussion on the potential theoretical tools for studying the role of augmentation on the SSL AD?
2. The observations may lack a clear outline and summarization to draw an in-depth conclusion for a better understanding of the major significance of the comprehensive study for SSL AD. Maybe it can be reorganized to provide a single section to clearly present the overall analytical results.

---

> ### Author Response · Authors · 2023-06-22
> **Response to Reviewer aT5C**
>
> We appreciate the reviewer's constructive feedback on the summary of observations and future research directions. We have addressed each point individually.
>
> ### Requested change 1
>
> > Summarize all the observations for a clear presentation and provide the corresponding discussion if there is some relationship among all those empirical results.
>
> We created a new section (Sec. 6) and summarized all our observations in the first part of that section (Sec. 6.1). We categorized our observations into three groups: (a) the success and failure of SSAD, (b) the empirical performance and bias of SSAD, and (c) case studies on individual samples. We aimed to present our observations in plain English so that they provide useful insights for future research easily.
>
> ### Requested change 2
>
> > Discuss more about future suggestions after presenting the major insights of the comprehensive study, which may help to enhance the significance of this research study.
>
> Following the summary of our observations in Sec. 6.1, we added a discussion on future research directions in SSAD in Sec. 6.2, based on our experimental results. The main idea is to devise an unsupervised validation loss that allows one to estimate the alignment between augmentation and anomalies in the low-dimensional embedding space, without requiring any labels. This validation loss is enabled in *transductive learning*, where unlabeled test data are available at training time. Our experimental results on data embeddings support this approach (Figs. 8 and 9).

---

### Review · Reviewer_hH9B · 2023-05-29

**Summary Of Contributions:**

The paper conducts a comprehensive examination of the impact of data augmentation in self-supervised learning (SSL)-based anomaly detection (SSAD) on image data. It found that the alignment between the data augmentation mechanism and the anomaly-generating mechanism is critical for the success of SSAD. With poor alignment, SSL may even degrade accuracy. The paper highlights that data augmentation remains a vital hyperparameter for SSAD, necessitating careful selection to unlock its full potential. It presents a meticulous study, employing 3 different types of SSAD models across 420 different anomaly detection tasks. It also cautions that unthoughtful implementation of SSL might lead to biased detection outcomes across anomaly types. This research marks the first meta-analysis study on SSAD and emphasizes the necessity for mindful implementation of SSL for accurate and fair anomaly detection.

**Audience:**

Yes

**Broader Impact Concerns:**

The paper presents potential broader impact concerns:

1. **Misuse**: The developed anomaly detection technique could be used maliciously, such as in invading privacy or discrimination.

2. **Bias**: If the training data is biased, the anomalies detected could lead to discriminatory outcomes.



**Claims And Evidence:**

Yes

**Requested Changes:**

The paper could be improved in a couple of ways:

1. **Dataset Size**: All the datasets used in the study are relatively small. To draw more general conclusions, it would be beneficial to include a larger dataset, such as ImageNet.

2. **Problem Solving**: The observations made in the paper are sound. However, it would enhance the value of the study if the authors could discuss possible solutions to the identified problems and propose clear, actionable strategies.


**Strengths And Weaknesses:**

## Strengths:

**Comprehensive Study**: The study conducted a large number of experiments across a variety of tasks and models, leading to robust and reliable insights about the impact of data augmentation on self-supervised anomaly detection (SSAD).

**In-Depth Analysis**: The study goes beyond simply reporting performance measures. It provides an in-depth analysis of the reasons behind the success or failure of SSAD, especially highlighting the critical role of data augmentation.

**Essential Insights**: The study underlines the critical importance of alignment between pseudo and true anomalies and shows how poor alignment can harm performance or lead to biased error distribution.

## Weaknesses:

**Hyperparameter Selection**: While the study highlights the critical role of data augmentation in SSL for AD, it doesn't provide clear guidance on how to select these hyperparameters effectively, which could be a challenging task, especially in unsupervised learning settings.

**Limited to Image Data**: The study primarily focuses on image-based SSAD. While the findings might be generalizable to other types of data, explicit verification is necessary. It might not provide much guidance for researchers working with non-image data.

**No New Model**: The research doesn't propose any new model or technique to improve upon existing SSAD methods. Instead, it emphasizes the importance of considering data augmentation as a hyperparameter, which might already be a known factor for researchers in the field.

**No Clear Solution**: The paper highlights issues with hyperparameter selection in unsupervised anomaly detection but doesn't provide a clear solution for resolving them. This might leave readers unsure about how to apply the insights from the study in practice.

---

> ### Author Response · Authors · 2023-06-22
> **Response to Reviewer hH9B**
>
> We appreciate the reviewer's constructive feedback on dataset size and problem-solving strategies. We have addressed each point individually.
>
> ### Requested change 1
>
> > **Dataset Size**: All the datasets used in the study are relatively small. To draw more general conclusions, it would be beneficial to include a larger dataset, such as ImageNet.
>
> While we understand the reviewer's concern about the size of the datasets, our work prioritizes the diversity of experimental settings (i.e., 3 different models on 420 different AD tasks) over the size of datasets. Our experiments show that our findings and observations are consistent across all four datasets, especially on all three detector models which have different motivations and objective functions. In addition, most existing works on image anomaly detection (Golan & El-Yaniv, 2018; Bergman & Hoshen, 2020; Li et al., 2021) use the same datasets as in our work, such as CIFAR-10 or FashionMNIST. A potential way to further improve the generalizability of our work can be to conduct experiments on non-image datasets, such as time series or tabular data, which we leave as a future work.
>
> ### Requested change 2
>
> > **Problem Solving**: The observations made in the paper are sound. However, it would enhance the value of the study if the authors could discuss possible solutions to the identified problems and propose clear, actionable strategies.
>
> We concur that offering insights and suggestions for future research are the core values of meta-analysis studies like ours. Hence, we added a new section (Sec. 6), summarizing all our findings and observations (Sec. 6.1) and introducing new research directions based on our experimental results (Sec. 6.2). The main idea is to devise an unsupervised validation loss that allows for estimating the alignment between augmentation and anomalies in the low-dimensional embedding space, without requiring any labels. This is facilitated in *transductive learning*, where unlabeled test data are available at training time. Our experimental results on data embeddings support this approach (Figs. 8 and 9).

---

### Review · Reviewer_SLGE · 2023-05-31

**Summary Of Contributions:**

the authors of this article delve into the intricate role of data augmentation in the realm of self-supervised learning (ssl) applied to anomaly detection (ad), especially when handling image data. this technology offers a cost-effective solution for generating supervisory signals, making it a key tool for tasks where anomalies are scant or even non-existent.

although recent studies have acknowledged the substantial effect of the type of augmentation on accuracy, the authors strive to expand this understanding by putting image-based ssl for ad under more extensive scrutiny. their research, carried out on three different detection models and across 420 anomaly detection tasks, presents exhaustive numerical and visual proofs.

the crux of their findings reveals the success of ssl-based ad heavily hinges on the alignment of the data augmentation and anomaly-generating mechanism. when this alignment is absent, ssl might even hamper accuracy.

**Audience:**

Yes

**Broader Impact Concerns:**

none as of this time

**Claims And Evidence:**

Yes

**Requested Changes:**

## conceptual
1. p.6 under "evaluation" it says "aug and gen are perfectly aligned if they are the same function, and still highly aligned if they are in the same family, such as aug:=Rotate and gen:=Flip – both of which are geometric augmentations." would it not make more sense to determine "families" based on the representation of the (augmented) data in the feature space of the trained ad as this is the notion of similarity that is ultimately of interest (i.e. similarity of the (augmented) data as represented/viewed by the model of interest). i do not expect the authors to adjust their entire testbed as there is value and insights already there. but im curious about a discussion of this conceptual nuance.
1. similarly, in-the-wild evaluation with mmd on resnet50 features presents a chicken-egg problem. alignment between aug data and normal data is conditioned on resnet50 feature space. which means downstream conclusions based on this alignment score are conditioned on resnet50 feature space. how does this influence the observations in the experiments? how could alignment be more objectively determined? for example, could not the feature space of the ad model in question be used for mmd calculation?
1. a visual abstract (see details under weaknesses)

## related work
1. choice of optimal augmentation in ssl there exists plenty of work, a good reference to discuss is the infomin paper https://proceedings.neurips.cc/paper/2020/hash/4c2e5eaae9152079b9e95845750bb9ab-Abstract.html
1. the impact of "semantic" alignment between datasets and downstream model performance has also been investigated for example in semi-supervised learning for example https://www.computer.org/csdl/journal/ai/2023/02/09762063/1CMrqSyVYTC
1. generative models have also been thoroughly investigated, leading to surprising observations where supposedly ood data has higher likelihood under the generative model than the training data itself (see eric nalisnick's paper https://openreview.net/forum?id=H1xwNhCcYm figure 2b). this has also been treated in more generality as relating to the counterintuitive behavior of scoring functions for generative models when it comes to semantic expectations of humans (see for example lucas theis' great work https://arxiv.org/abs/1511.01844)
## typos, notation, language, presentation
1. provide tables from ding et al mentioned at bottom of p.2 in appendices for easier reference
1. in sentence "whereas use ImageNet-22K as OE data (superset of ImageNet-1K) to evaluate on ImageNet" on p.3 missing subject
1. incomplete mnist citation -> not to dunk on yann lecun and gang but: the mnist dataset was collected, curated and prepared by wilson, wilkinson, garris and colleagues*. yann and colleagues popularized the dataset later.
1. wording in finding 1 "(ii) faug impairs f if the alignment between aug and gen is poor." seems off. shouldnt it read "(ii) aug impairs f if the alignment between aug and gen is poor." or "(ii) f surpasses faug if the alignment between aug and gen is poor."
1. p.9 observation 5, notation "DAEfaug" looks a bit odd

\*
CL Wilson and MD Garris. Handprinted character database. technical report special database 1. Technical
report, National Institute of Standards and Technology, 1990. 2

MD Garris and RA Wilkinson. Handwritten segmented characters database. technical report special database 3.
Technical report, National Institute of Standards and Technology, 1992.

Michael Garris. Design, collection, and analysis of handwriting sample image databases. (31), 1994-08-10 1994.
URL https://tsapps.nist.gov/publication/get_pdf.cfm?pub_id=906483.

## criticality of changes
id like to give authors opportunity to respond to my questions to clear up misunderstandings. overall, the visual abstract would be strongest addition in my point of view. discussing the conceptual concerns and related work would further round out the manuscript.

**Strengths And Weaknesses:**

## strengths
1. thorough and deep treatment of an overlooked issue: the impact of the choice of augmentation on the ssl ad performance
1. excellent exposition and contextualization (abe, steinwart, theiler references in section 1) of a complicated concepts (controlled testbed, curse of obviousness -> see watts reference in footnote 2)
1. thorough quantitative and qualitative experiments
1. competent, structured and reader-friendly presentation of results (with finding and observation environments in section 5)

## weaknesses
1. given the richness in concepts, experiments and results of the manuscript as well as the ambition of the authors to consolidate/sort the field, a visual abstract at the beginning of the paper would be very helpful. it would provide others with a blueprint on the experimental design (i.e. the testbed) as well as an illustration of the relationship between [gen, in-the-wild, aug]X[datasets used in testbed (e.g. svhn, mnist, etc.)]X[ads used]X[main results]
1. as a paper that aims to connect the dots on things ssl, unsupervised learning and ad, the discussion could be enriched further with concepts and findings from existing work work

---

> ### Author Response · Authors · 2023-06-22
> **Response to Reviewer SLGE**
>
> We appreciate the reviewer's detailed and constructive feedback on various aspects of our work: a visual abstract, conceptual concerns, related works, and presentation. We have addressed the points raised individually.
>
> ### Requested change 1
>
> > p.6 under "evaluation" it says "aug and gen are perfectly aligned if they are the same function, and still highly aligned if they are in the same family, such as aug:=Rotate and gen:=Flip – both of which are geometric augmentations." would it not make more sense to determine "families" based on the representation of the (augmented) data in the feature space of the trained ad as this is the notion of similarity that is ultimately of interest (i.e. similarity of the (augmented) data as represented/viewed by the model of interest). i do not expect the authors to adjust their entire testbed as there is value and insights already there. but im curious about a discussion of this conceptual nuance.
>
> We acknowledge that defining alignment based on the feature space of the AD model might be more sensible, considering this is the similarity notion ultimately of interest. However, our focus on the functional family is due to its intuitive and model-agnostic nature. As the first systematic study on the alignment of augmentation in SSAD, we aim to concentrate on a high-level idea that is generalizable across various settings, rather than proposing model-specific solutions.
>
> Nevertheless, we concur that the distribution in a low-dimensional feature (or embedding) space can effectively quantify the degree of alignment between any augmentation and anomaly-generating functions. Our experimental results also show that the distribution of embeddings can be a key idea to estimate the alignment even without considering the functional form of augmentation (Figs. 8 and 9). We believe this insight not only provides a formal definition of alignment but can also guide future directions of augmentation tuning in SSAD. We discuss this in more detail in Sec. 6.2.
>
> ### Requested change 2
>
> > similarly, in-the-wild evaluation with mmd on resnet50 features presents a chicken-egg problem. alignment between aug data and normal data is conditioned on resnet50 feature space. which means downstream conclusions based on this alignment score are conditioned on resnet50 feature space. how does this influence the observations in the experiments? how could alignment be more objectively determined? for example, could not the feature space of the ad model in question be used for mmd calculation?
>
> We concur that using MMD alignment based on ResNet50 features isn't an ideal way to quantify alignment. We opted for it as it isn't specific to a detector model choice, and ResNet50 is a widely accepted model we can use without resorting to several pre-trained models, which we prefer to avoid in unsupervised settings. However, in accordance with our response to Requested Change 1, we agree that MMD (or a similar distributional distance) in the feature space of the AD model itself might better quantify alignment. Due to space limitations in the revised paper, we have decided to move the MMD-based alignment discussion to the appendix.
>
> ### Requested change 3
>
> > a visual abstract (see details under weaknesses)
>
> We have included a visual abstract of our work as Fig. 1, comprising three components: (a) the operation of SSAD, (b) our two testbeds, and (c) our primary finding concerning the alignment between augmentation and anomalies.
>
> ### Related work
>
> We have bolstered our discussion on related works, creating a new section (Sec. 2.2) that discusses self-supervised learning in a more comprehensive manner, including references to Theis et al. (2015), Nalisnick et al. (2018), Tian et al. (2020), and Calderon-Ramirez et al. (2022).
>
> ### Typos, notation, language, presentation
>
> We have made improvements to the notation and language based on your suggestions:
>
> 1. We think that presenting the experimental results from Ding et al. (2022) directly in our paper is not easy, since it requires us to provide sufficient background information to make them understandable on their own. Instead, we have changed the sentence mentioning Ding et al. (2022) to make it more understandable without actually seeing their results.
> 2. We fixed the typo.
> 3. We additionally cited (Garris, 1994) for MNIST.
> 4. We changed it into “$f$ surpasses $f_\mathrm{aug}$”.
> 5. We used the terms “DAE f_aug” and “AE f” in Observations 5 - 7 to specifically refer to a DAE-based detector network. We are willing to improve the wording based on your further suggestion.

---

> > ### Comment · Reviewer_SLGE · 2023-06-22
> > **response to manuscript updates**
> >
> > dear authors,
> >
> > thank you for the detailed response and updated manuscript.
> >
> > i particularly appreciate the great visual abstract, the expanded related work as well as the condensed findings and the additional mmd results in the appendices.
> >
> > i also appreciate your clarification to my remarks regarding the choice of the feature space to determine alignment. the design constraints you chose (model agnostic) sound reasonable to me.
> >
> > finally, only one minor note on notation: in figure 17 it says "in Obs. ??".
> >
> > in good spirits,
> > reviewer slge

---

### Author Response · Authors · 2023-06-22
**Summary of Changes**

We extend our gratitude to all reviewers for their insightful comments. We have carefully revised our paper and uploaded the updated version to the system, ensuring the changes fall within the 12-page limit. The modifications are highlighted in blue. Here are the primary amendments we have made:

- **Visual abstract:** We included a visual abstract (Fig. 1), which demonstrates how self-supervised anomaly detection (SSAD) operates, presents our two testbeds, and our main finding about the alignment between augmentation and anomalies.
- **Related works:** We enhanced our discussion on related works, generating a new section (Sec. 2.2) that introduces and deliberates on self-supervised learning in a broader context.
- **Summary of findings:** We incorporated a summary of all our findings and observations in Sec. 6.1, categorizing them into three groups: (1) the success and failure of SSAD, (2) the empirical performance and bias of SSAD, and (3) case studies.
- **Future directions:** We proposed future research directions in SSAD based on our findings and observations. The central idea is to explore *transductive learning*, where unlabeled test data can assist in minimizing an *unsupervised validation loss* for augmentation tuning.

---

### Decision · Action_Editors · 2023-07-07

**Recommendation:** Accept as is

**Comment:**

The paper offers a comprehensive study on data augmentation in anomaly detection, and the authors have addressed various suggestions put forth by the reviewers. These include the inclusion of a visual abstract, expansion of the related work section, and the provision of additional results. While one reviewer expressed concern about the lack of a definitive resolution to the observed distribution shift, it is important to note that this paper's primary contribution lies in its systematic empirical study. The formulation of a concrete algorithmic solution can be considered as potential future work in this area.

**Audience:**

The research explores the intricate role of data augmentation in SSL for AD, providing insights into the effectiveness and alignment of augmentation techniques for anomaly detection. Given the increasing importance of SSL and AD in various applications, the findings presented in the paper can contribute to the understanding and improvement of anomaly detection techniques.

**Claims And Evidence:**

The authors undertook a thorough investigation into the influence of data augmentation on anomaly detection, focusing specifically on self-supervised learning (SSL) applied to anomaly detection (AD) for image data. Their study encompassed extensive numerical and visual evidence derived from research conducted across 420 anomaly detection tasks, utilizing three distinct detection models. The findings strongly indicate that the effectiveness of SSL-based AD critically depends on aligning the mechanisms of data augmentation and anomaly generation.